# Towards Achieving Optimal Strong Regret and Constraint Violation via Computationally Efficient Model-free RL

**Xiyue Peng**[1]   **Lingkai Zu**[1]   **Ziyu Shao**[1]   **Xin Liu**[1]

## Abstract

We study episodic constrained Markov decision processes (CMDPs) with linear function approximation, where the goal is to achieve strong regret and constraint violation guarantees without allowing error cancellations. Unlike the existing work, which focuses on either tabular CMDP or model-based reinforcement learning methods. We propose a model-free policy APMPO that achieves near-optimal $\widetilde{O}(\sqrt{K})$ strong regret and strong constraint violation with Slater's condition (or strict feasibility assumption), where $K$ is the total number of episodes. It matches the best-known rates without requiring any prior knowledge of the feasibility gap reported in prior model-based work for tabular CMDPs. Besides, APMPO achieves $\widetilde{O}(K^{\frac{3}{4}})$ strong regret and $\widetilde{O}(K^{\frac{3}{4}})$ strong constraint violation without Slater's condition. To the best of our knowledge, this is the first sublinear result of CMDP w.r.t. the strong metrics without Slater's condition. APMPO achieves these results by a novel and adaptive design of a violation-aware penalty and learning rates to balance the strong regret and constraint violation, which is quite different from the (regularized) primal-dual methods imposing constraints via dual penalty in the literature. The experiments show APMPO significantly outperforms the strong baselines, which justify our design and theoretical performance.

## 1. Introduction

Reinforcement learning (RL) studies how an agent can learn a policy to optimize long-term performance through sequential interaction with an environment, which is commonly modeled as a Markov decision process (MDP). While classical RL focuses on maximizing cumulative rewards, many real-world applications impose additional requirements during the learning process, such as safety requirements for autonomous systems (Fisac et al., 2018; Ono et al., 2015; Garcia & Fernández, 2012), budget constraints in online advertising (Duan et al., 2025), fairness or service-level guarantees in online platforms (Abe et al., 2010). Such requirements are naturally captured by constrained Markov decision processes (CMDPs) (Altman, 1999), where the agent seeks to maximize expected cumulative reward while satisfying critical constraints on cumulative costs. In this paper, we focus on finite-horizon episodic CMDPs, in which the agent interacts with the environment over $K$ episodes, each consisting of $H$ steps.

Seminal results by Efroni et al. (2020) established comprehensive regret guarantees for LP-based, primal-dual, and dual reinforcement learning policies for CMDPs, inspiring many subsequent studies. Among these, primal-dual (or purely dual) RL policies are often preferred in practice over LP-based ones due to their computational efficiency and flexibility, especially in large-scale and deep RL applications (Tessler et al., 2019; Chow et al., 2018; Achiam et al., 2017). However, the theoretical guarantees for these primal-dual and dual RL policies typically pertain only to weak regret and constraint violation, which bound the cumulative sum of per-episode errors. As noted by Efroni et al. (2020), such weak metrics permit "error cancellation": large positive errors in some episodes can be offset by negative errors, still yielding a small cumulative value. Although analytically convenient, this property renders weak metrics unsuitable for safety-critical applications, because a policy with sublinear weak violations may still frequently and unsafely violate constraints during learning. While Efroni et al. (2020) also introduced strong (or strict) regret and constraint violation metrics, which accumulate only positive per-episode errors, obtaining such guarantees is notably difficult within the standard primal-dual (or dual) framework, due to limitations inherent in the convex-analytic analyses of these methods (Efroni et al., 2020; Beck, 2017).

While the LP-based method of Efroni et al. (2020) first demonstrated the achievability of optimal $\tilde{O}(\sqrt{K})$ strong regret and strong constraint violation, it again requires solving

[1]School of Information Science and Technology, ShanghaiTech University, Shanghai, China. Correspondence to: Xin Liu <liuxin7@shanghaitech.edu.cn>.

*Proceedings of the 43rd International Conference on Machine Learning*, Seoul, South Korea. PMLR 306, 2026. Copyright 2026 by the author(s).

| Method | Strong Regret/Violation (Slater's Condition) | Strong Regret/Violation (Slater's Condition Free) | Model Free/ Function Approximation |
|---|---|---|---|
| **APMPO** | $\tilde{O}(\frac{\sqrt{K}}{\rho^2})$, $\tilde{O}(\frac{\sqrt{K}}{\rho})$ ($\rho$ is unknown) 
 $\tilde{O}(\frac{\sqrt{K}}{\rho})$, $\tilde{O}(\sqrt{K})$ ($\rho$ is known) | $\tilde{O}(K^{3/4})$, $\tilde{O}(K^{3/4})$ | ✓, linear |
| (Stradi et al., 2025a) | $\tilde{O}(\frac{\sqrt{K}}{\rho})$, $\tilde{O}(\sqrt{K})$ | ✗ | ✗, tabular |
| (Müller et al., 2024) | $\tilde{O}(K^{0.92})$, $\tilde{O}(K^{0.92})$ | ✗ | ✗, tabular |

*Table 1.* **Comparison of Algorithms for Strong Regret and Constraint Violation.** APMPO is the first algorithm to achieve sublinear strong regret and constraint violation without Slater's condition or knowledge of the Slater constant $\rho$, which is required by Müller et al. (2024) and Stradi et al. (2025a). When Slater's condition holds, APMPO matches the best-known rates of Stradi et al. (2025a). Moreover, APMPO is model-free with linear function approximation, whereas prior results rely on model-based methods in tabular settings.

a linear program over occupancy measures in each episode, as discussed above. This highlights a recurring tension between computational efficiency in practice and the strong theoretical guarantees of regret and constraint violation. Furthermore, although the primal-dual method of Ghosh et al. (2024) attains sublinear weak regret alongside strong constraint violation, it incurs exponential-time complexity due to its search for suitable dual variables per episode. Two recent works have sought to overcome this tension by pursuing more computationally efficient primal-dual methods that can deliver strict guarantees. Müller et al. (2024) obtains suboptimal $\tilde{O}(K^{0.93})$ strong regret and violation by employing a regularized framework. Stradi et al. (2025a) later improved the result to $\tilde{O}(\sqrt{K})$ via a switching-type dual update. However, both algorithms are model-based and limited to the tabular setting, which can not scale to problems with large or continuous state spaces. Therefore, it motivates the following fundamental question:

*Can we design a computationally efficient, model-free reinforcement learning algorithm that achieves near-optimal strong regret and strong constraint violation in CMDPs with function approximation?*

We provide a positive answer to this question. We study episodic CMDPs with linear function approximation and propose a model-free algorithm APMPO that achieves near-optimal strong regret and constraint violation. Unlike classical primal–dual approaches that impose constraints via dual penalties, APMPO employs a violation-aware adaptive penalty mechanism, along with carefully designed learning rates, to balance reward optimization and constraint satisfaction under strong performance criteria. This design avoids the oscillations inherent in standard primal–dual methods (Müller et al., 2024) and enables sharp control of strong metrics. As shown in Table 1, we would like to emphasize following advantages of APMPO:

- With Slater's condition, APMPO achieves $\widetilde{O}(\sqrt{K})$ strong regret and constraint violation without prior knowledge of the Slater constant $\rho$, and matches the best-known rates as

tabular setting in Stradi et al. (2025a) when $\rho$ is known.
- Without Slater's condition, APMPO is the first algorithm to attain sublinear strong regret and strong constraint violation without assuming Slater's condition, via a model-free design with linear function approximation.

**Conflict of Interest Disclosure.** The authors declare no financial conflicts of interest related to this work.

## 2. Related Works

CMDPs have been studied through two primary algorithmic lenses. One line of work, including Altman (1999); Efroni et al. (2020); Singh et al. (2020); Kalagarla et al. (2021); Liu et al. (2021a); Brantley et al. (2020); Bura et al. (2022), adopts a model-based, linear programming (LP) approach. These methods typically first learn a model of the transition dynamics and then solve the corresponding LP over occupancy measures. While foundational, this approach often faces scalability challenges in large state spaces. In contrast, another major direction utilizes primal-dual or dual optimization frameworks, as seen in works such as Paternain et al. (2019; 2022); Qiu et al. (2020); Chen et al. (2021); Liu et al. (2021b); Bai et al. (2022); Ding & Jovanović (2022); Ghosh et al. (2022); Tian et al. (2024); Ding et al. (2025); Liu et al. (2025); Zuo & He (2025); Ding et al. (2020). These methods are often more flexible and have been applied to various settings, including those with continuous state spaces, by directly optimizing the Lagrangian without explicitly solving a large-scale LP. A notable common limitation of these lines of work, however, is that they generally provide only weak regret and constraint violation guarantees, which are insufficient for online performance assessment in high-stakes or safety-critical environments.

A related but distinct line of work studies safe exploration and stringent constraint control in constrained sequential decision-making (Liu et al., 2021c; Bura et al., 2022; Yu et al., 2024; Ni & Kamgarpour, 2023). These works pursue an objective different from ours and mostly focus on

standard regret or soft notions of constraint violation. In particular, Liu et al. (2021c) study a constrained bandit setting with soft cumulative constraint violation, which may vanish after a finite period but can still allow large hard violations during learning. The other works provide safe exploration guarantees in CMDPs or constrained RL, but they are mainly developed for tabular or model-based settings and rely on the availability of a strictly safe baseline policy, a known safe set, or other prior safety knowledge. As noted by Ni & Kamgarpour (2023), such prior information is natural in safe exploration because, without a safe initial policy, safety from the outset cannot be guaranteed. In contrast, we study strong regret and strong cumulative constraint violation for model-free CMDPs with linear function approximation, without assuming access to a strictly safe baseline policy or even its existence in the Slater-free setting. Thus, our results characterize the trade-off between exploration and constraint satisfaction without requiring per-episode zero violation. We also note a related line of work on online convex optimization with long-term, hard, or adversarial constraints (Yuan & Lamperski, 2018; Yi et al., 2021; Guo et al., 2022; Sinha & Vaze). These works study regret together with non-canceling notions of cumulative constraint violation in convex online learning, and are conceptually related to our strong-violation metric. However, they do not involve sequential decision-making, unknown transition dynamics, or value-function approximation.

We next discuss the most directly related works on strong regret and strong constraint violation. The LP-based method OptCMDP by Efroni et al. (2020) achieves the optimal $\tilde{O}(\sqrt{K})$ rate for both strong regret and strong violation by solving a linear program over state-action occupancy measures. However, the LP variable dimension scales with the size of the state space, making this approach difficult to apply in large-scale or continuous-state problems. In a different direction, Ghosh et al. (2024) employ a primal–dual approach to obtain sublinear weak regret together with strong constraint violation, but their algorithm requires an exponential-time search for suitable dual variables in each episode. More recent advances seek computationally efficient primal–dual methods with strong guarantees. Müller et al. (2024) obtain sublinear strong regret and violation via a regularized dual formulation, but their bounds scale as $\tilde{O}(K^{0.92})$. Stradi et al. (2025a) improve the dependence on $K$ to the optimal $\tilde{O}(\sqrt{K})$ rate through a switching-type dual update with optimistic value estimation. Nevertheless, both methods are model-based and restricted to tabular CMDPs. They also require prior knowledge of the Slater constant for parameter tuning.

Finally, we discuss two related works on adversarial MDPs with stochastic hard constraints and online optimization with long-term constraints, namely Stradi et al. (2025b) and Castiglioni et al. (2022). These works are related to ours

through their use of Slater-type feasibility conditions and feasibility-gap information. However, their treatment of the Slater constant $\rho$ differs from ours. Both works involve a dedicated stage or procedure to estimate $\rho$, whereas our approach is a one-stage adaptive method that does not require prior knowledge of $\rho$. Instead, it updates the penalty online according to the predicted constraint shortfall. Our setting is also different: we study model-free CMDPs with linear function approximation under strong regret and strong constraint-violation metrics. As summarized in Table 1, when $\rho$ is known and the penalty parameter is tuned accordingly, our method achieves even stronger guarantees. The techniques developed by Stradi et al. (2025b) may be complementary to ours and could potentially be incorporated into our framework.

Taken together, these comparisons motivate `APMPO`, a computationally efficient model-free algorithm for CMDPs with linear function approximation. By combining optimistic value estimation with an adaptive episode-wise penalty, our method avoids solving occupancy-measure LPs, performing exponential dual search, or running a separate procedure to estimate the Slater constant. It provides near-optimal dependence on $K$ for both strong regret and strong constraint violation.

## 3. Problem Formulation

We consider a finite-horizon episodic CMDP, defined by $(\mathcal{S}, \mathcal{A}, H, \{P_h\}_{h=1}^H, \{r_h\}_{h=1}^H, \{g_h\}_{h=1}^H)$, where $\mathcal{S}$ denotes the state space (potentially infinite), $\mathcal{A}$ is the finite action set, and $H$ is the episode horizon. At each step $h \in [H]$ ($[H] = \{1, \cdots, H\}$) of an episode, the agent observes a state $x_h \in \mathcal{S}$, selects an action $a_h \in \mathcal{A}$, receives a reward $r_h(x_h, a_h)$, and incurs a utility $g_h(x_h, a_h)$. The rewards and utilities are drawn from i.i.d. distributions, and we use the term *utility* to emphasize that the constraint for is lower bounded by a threshold in the CMDP problem. The agent then transitions to the next state according to the kernel, i.e. $x_{h+1} \sim P_h(\cdot \mid x_h, a_h)$. The reward, utility, and transition kernel are initially unknown to the agent. The agent interaction lasts for $K$ episodes. For a simple exposition, we assume a fixed initial state $x_1$ across all episodes.

A policy is denoted by $\pi = \{\pi_h\}_{h=1}^H$, where each step $\pi_h : \mathcal{S} \to \Delta(\mathcal{A})$. For any policy $\pi$, let $V_{u,h}^\pi(x)$ and $Q_{u,h}^\pi(x, a)$ with $u \in \{r, g\}$ denote the value functions of reward or utility from step $h$ when starting from state $x$.

$$V_{u,h}^\pi(x) := \mathbb{E}\left[\sum_{t=h}^H u_t(x_t, a_t) \,\middle|\, x_h = x\right],$$

$$Q_{u,h}^\pi(x, a) := \mathbb{E}\left[\sum_{t=h}^H u_t(x_t, a_t) \,\middle|\, x_h = x, a_h = a\right],$$

where the expectation is taken with respect to the trajectory induced by policy $\pi$ and the environment. The (action) value functions of reward or utility satisfy the Bellman equations

$$Q_{u,h}^{\pi}(x,a) \triangleq u_h(x,a) + \mathbb{E}_{x' \sim P_h(\cdot|x,a)}\left[V_{u,h+1}^{\pi}(x')\right],$$

where we have the convention that $V_{u,H+1}^{\pi}(\cdot) \equiv 0$.

The agent's objective is to maximize expected cumulative reward while satisfying the constraint with threshold $b$:

$$\max_{\pi} V_{r,1}^{\pi}(x_1) \quad \text{s.t. } V_{g,1}^{\pi}(x_1) \geq b. \quad (1)$$

Let $\pi^{\star}$ denote an optimal policy to (1). For simplicity, we present the problem with a single utility/constraint, and our results can be readily extended to the case with multiple utilities/constraints.

**Performance metrics:** During each episode $k \in [K]$, the agent selects a policy $\pi_k$ before the episode starts, executes $\pi_k$ for one full episode. The conventional (weak) regret and constraint violation are defined as follows

$$\mathcal{R}(K) = \sum_{k=1}^{K}\left(V_{r,1}^{\pi^{\star}}(x_1) - V_{r,1}^{\pi_k}(x_1)\right),$$

$$\mathcal{V}(K) = \sum_{k=1}^{K}\left(b - V_{g,1}^{\pi_k}(x_1)\right).$$

We further define *strong* regret and *strong* constraint violation, which do not allow error cancellations across episodes.

$$\mathcal{R}_+(K) = \sum_{k=1}^{K}\left(V_{r,1}^{\pi^{\star}}(x_1) - V_{r,1}^{\pi_k}(x_1)\right)_+,$$

$$\mathcal{V}_+(K) = \sum_{k=1}^{K}\left(b - V_{g,1}^{\pi_k}(x_1)\right)_+.$$

Unlike weak regret and violation, these metrics penalize every episode in which the selected policy is suboptimal or infeasible during the learning process, making them particularly suitable for safety-critical applications.

**Linear function approximation:** To address large or continuous state spaces, we consider linear function approximation for the environment (Jin et al., 2020; Ghosh et al., 2024).

**Assumption 3.1** (Linear CMDP)**.** There exists a known feature map $\phi : \mathcal{S} \times \mathcal{A} \to \mathbb{R}^d$ such that for all $h \in [H]$,

$$r_h(x,a) = \langle \phi(x,a), \theta_{r,h} \rangle,$$
$$g_h(x,a) = \langle \phi(x,a), \theta_{g,h} \rangle,$$
$$P_h(\cdot \mid x,a) = \langle \phi(x,a), \mu_h(\cdot) \rangle,$$

where $\theta_{r,h}, \theta_{g,h} \in \mathbb{R}^d$ are unknown parameter vectors, and $\mu_h(\cdot)$ is a vector of signed measures over $\mathcal{S}$. Further, we assume for any $(x,a,h)$, $\|\phi(x,a)\|_2 \leq 1$, $\|\theta_{r,h}\|_2 \leq \sqrt{d}$, $\|\theta_{g,h}\|_2 \leq \sqrt{d}$, and $\|\mu_h(\mathcal{S})\|_2 \leq \sqrt{d}$.

According to Proposition 2.3 of Jin et al. (2020), Assumption 3.1 implies the action-value functions admit a linear representation in the feature map. That is, for any policy $\pi$, there exists a set of weights $\{w_{u,h}^{\pi}\}_{h=1}^{H}$ such that

$$Q_{u,h}^{\pi}(x,a) = \langle \phi(x,a), w_{u,h}^{\pi} \rangle, \ u \in \{r,g\}.$$

Linear MDPs provide a flexible and expressive framework that has proven effective in practice and influential in theory. It builds a principled foundation for analyzing reinforcement learning with function approximation in large-scale or infinite-state spaces. Note tabular MDPs arise as a special case of linear MDPs (Jin et al., 2020).

# 4. Adaptive Penalty Matching Policy Optimization

In this section, we present **Adaptive Penalty Matching Policy Optimization** (APMPO), a model-free algorithm for episodic linear CMDPs. APMPO integrates optimism in value estimation with a violation-adaptive penalty design and mirror-descent-based policy optimization, enabling efficient exploration while explicitly controlling *strong* regret and constraint violations across episodes.

---

**Adaptive Penalty Matching Policy Optimization**

---

**Initialization:** Initialize a policy $\pi_1 = \{\pi_{1,h}\}_{h=1}^{H}$, set a base penalty parameter $\lambda > 0$, and initialize data matrices $\Lambda_{0,h} = I_d$ for all $h \in [H]$.

**For episodes** $k = 1, \ldots, K$**:**

- **Observe:** The initial state $x_1^k$.
- **Policy execution:** Execute policy $\pi_k$ for one episode and observe the trajectory

$$(x_1^k, a_1^k, r_1^k, g_1^k, \ldots, x_H^k, a_H^k, r_H^k, g_H^k).$$

- **Optimistic value function estimation:** For each step $h = H, \ldots, 1$, update the data matrix

$$\Lambda_{k,h} = \Lambda_{k-1,h} + \phi(x_h^k, a_h^k)\phi(x_h^k, a_h^k)^{\top},$$

and compute the least-squares estimates

$$\hat{w}_{r,h}^k = \Lambda_{k,h}^{-1} \sum_{\tau=1}^{k} \phi(x_h^{\tau}, a_h^{\tau})\left(r_h^{\tau} + \hat{V}_{r,h+1}^k(x_{h+1}^{\tau})\right),$$

$$\hat{w}_{g,h}^k = \Lambda_{k,h}^{-1} \sum_{\tau=1}^{k} \phi(x_h^{\tau}, a_h^{\tau})\left(g_h^{\tau} + \hat{V}_{g,h+1}^k(x_{h+1}^{\tau})\right).$$

Define the optimistic action–value estimates

$$\hat{Q}_{r,h}^k(x,a) = \langle \hat{w}_{r,h}^k, \phi(x,a) \rangle + \beta_r^k \|\phi(x,a)\|_{\Lambda_{k,h}^{-1}},$$

$$\hat{Q}_{g,h}^k(x,a) = \langle \hat{w}_{g,h}^k, \phi(x,a) \rangle + \beta_g^k \|\phi(x,a)\|_{\Lambda_{k,h}^{-1}},$$

and the corresponding value functions

$$\hat{V}_{u,h}^{k}(x) = \sum_{a \in \mathcal{A}} \pi_{k,h}(a|x) \hat{Q}_{u,h}^{k}(x,a), \ \ u \in \{r,g\}.$$

- **Violation-adaptive penalty and learning rate:**

$$\eta_k = \lambda \left(b - \hat{V}_{g,1}^{k}(x_1^k)\right)_+, \ \alpha_k = \sqrt{\sum_{s=1}^{k}(1+\eta_s^2)}.$$

- **Policy optimization (mirror descent):** For every state $x$ and step $h$, define its penalty matching value function

$$F(x,\cdot) = \hat{Q}_{r,h}^{k}(x,\cdot) + \eta_k \hat{Q}_{g,h}^{k}(x,\cdot).$$

We update the policy $\pi_{k+1,h}(\cdot|x)$ by minimizing the strong violation regularized value functions

$$\langle F(x,\cdot), \pi_{k,h}(\cdot|x) - \pi(\cdot|x)\rangle + \alpha_k \, \mathrm{KL}(\pi(\cdot|x) \| \pi_{k,h}(\cdot|x)).$$

APMPO employs a novel violation-aware adaptive penalty mechanism to balance reward optimization against constraint satisfaction under strict performance criteria, while carefully designed adaptive learning rates to trade-off exploration and exploitation. The framework consists of the following key components.

**Optimistic value function learning.** APMPO adopts an optimistic value-function learning strategy under linear function approximation. This construction follows the standard linear MDP paradigm (Jin et al., 2020) and provides tight uncertainty quantification. At each episode, we use previously collected trajectories to perform regularized least-squares regression, obtaining estimates of the action-value functions for both the reward and constraint. We then add upper-confidence bonuses to obtain optimistic action-value estimates $\hat{Q}_{r,h}^{k}(x,a)$ and $\hat{Q}_{g,h}^{k}(x,a)$, which induce the corresponding value functions $\hat{V}_{r,h}^{k}(x)$ and $\hat{V}_{g,h}^{k}(x)$.

**Violation-aware Policy optimization.** APMPO directly integrates violation information into the "primal" policy update. This design allows the policy to react immediately to predicted constraint violations while preserving stability and computational efficiency. We highlight two key design components in APMPO.

**1) Adaptive penalty surrogate function.** At the beginning of each episode, the algorithm uses optimistic value estimates to predict whether the constraint will be violated. If the predicted constraint value falls below the threshold, a penalty is activated and scaled proportionally to the severity of the predicted violation; otherwise, no penalty is applied. As a result, constraint enforcement is neither delayed nor overly conservative. By embedding this adaptive penalty directly into the surrogate objective, APMPO ensures that

reward maximization and constraint satisfaction are jointly addressed within a single optimization target.

**2) Violation-aware policy mirror descent.** Given the penalty-matched surrogate objective, APMPO updates the policy via KL-regularized mirror descent at each state and step. Rather than selecting greedy actions, the policy update encourages smooth, entropy-aware evolution across episodes. This yields stable learning dynamics and avoids abrupt policy changes that can arise under aggressive penalty adjustments. To maintain robustness under varying penalty magnitudes, the learning rate is scaled adaptively across episodes. This prevents overly large updates when constraint violations are severe, ensuring that the policy remains stable even as constraint pressure increases.

**Comparison with primal–dual approaches.** Traditional primal–dual methods enforce constraints via iterative dual updates, which often induce oscillatory behavior and typically yield convergence guarantees only in an averaged or weak sense. Recent refinements alleviate these issues through regularization (Müller et al., 2024), aligned or switching dual updates (Stradi et al., 2025a), or exhaustive searches over feasible dual variables (Ghosh et al., 2024). However, these approaches either suffer from suboptimal performance, require additional feasibility information, or incur significantly increased algorithmic complexity. In contrast, APMPO incorporates constraint information through a directly computed, episode-wise penalty, eliminating the need for dual tuning, nested primal–dual updates, or explicit exploration over dual variables. This distinction is particularly important for strong metrics. Strong regret and strong constraint violation count positive per-episode errors and therefore do not permit cancellation across episodes, whereas standard primal–dual analyses are naturally aligned with averaged Lagrangian performance and weak violation guarantees. APMPO instead sets the penalty according to the predicted constraint shortfall in the current episode,

$$\eta_k = \lambda \left(b - \widehat{V}_{g,1}^{k}(x_1^k)\right)_+,$$

and matches this penalty with the same episode's policy update. As a result, the update is immediately responsive to predicted violations, while the analysis obtains a self-bounding structure in which the violation-dependent term appears both in the penalty magnitude and in the policy-optimization bound. This is the key mechanism that allows APMPO to control cumulative positive violations directly under strong metrics.

## 5. Theoretical Results

In this section, we present the theoretical performance of APMPO and sketch the proof. We begin by stating the main results on strong regret and strong constraint violation when Slater's condition holds, and then extend the results to

cases without Slater's condition. For simple exposition, the choices of hyperparameters in APMPO can be found in the proof.

## 5.1. Main Results with Slater's Condition

**CMDP with Slater's Condition.** Our analysis leverages the following feasibility assumption to achieve near-optimal results, as in the tabular CMDP literature (Stradi et al., 2025a).

**Assumption 5.1** (Slater's condition). *There exists a policy $\pi$ and a constant $\rho > 0$ such that $V_{g,1}^{\pi}(x_1) \geq b + \rho$.*

The constant $\rho$ is referred to as the *Slater's constant* or *feasibility gap*.

**Translating weak regret into strong regret.** Firstly, we present Theorem 5.2 to demonstrate that APMPO achieves weak regret and strong violation bounds when Slater's condition holds.

**Theorem 5.2.** *Given any $p \in (0, 1)$, APMPO achieves the following weak regret and strong constraint violation with a probability at least $1 - p$:*

$$\mathcal{R}(K) = \tilde{O}(\sqrt{KH^5} + \sqrt{d^3 H^3 K}),$$

$$\mathcal{V}_+(K) = \tilde{O}(\frac{\sqrt{KH^5}}{\rho} + \sqrt{d^3 H^3 K}).$$

*If the Slater's constant $\rho$ is known, there is a tighter strong regret and strong constraint violation bound with probability at least $1 - p$:*

$$\mathcal{R}(K) = \tilde{O}(\frac{\sqrt{KH^5}}{\rho} + \sqrt{d^3 H^3 K}),$$

$$\mathcal{V}_+(K) = \tilde{O}(\sqrt{KH^5} + \sqrt{d^3 H^3 K}).$$

*Remark* 5.3. Theorem 5.2 also yields a PAC-style sample-complexity guarantee for a final returned policy via the standard online-to-batch conversion (Jin et al., 2018). Specifically, after running APMPO for $K$ episodes, we return $\pi_{\text{out}} = \pi_k$, where $k$ is sampled uniformly from $[K]$. If $K$ is chosen such that $R(K) \leq c\epsilon K$ and $V_+(K) \leq c\epsilon K$ for a sufficiently small absolute constant $c$, then the returned policy satisfies the following PAC-style bounds

$$V_{r,1}^{\pi^*}(x_1) - V_{r,1}^{\pi_{\text{out}}}(x_1) \leq \epsilon, \qquad \left(b - V_{g,1}^{\pi_{\text{out}}}(x_1)\right)_+ \leq \epsilon$$

with at least constant probability over the random choice of $k$. For example, with known Slater's constant $\rho$, Theorem 5.2 implies an episode complexity of

$$\tilde{O}\left(\max\left\{\frac{H^5}{\rho^2\epsilon^2}, \frac{d^3 H^3}{\epsilon^2}\right\}\right),$$

or equivalently,

$$\tilde{O}\left(\max\left\{\frac{H^6}{\rho^2\epsilon^2}, \frac{d^3 H^4}{\epsilon^2}\right\}\right)$$

transition samples, since each episode consists of $H$ steps.

To translate the weak regret $\mathcal{R}(K)$ to the strong one $\mathcal{R}_+(K)$. We establish the following key lemma.

**Lemma 5.4.** *Suppose a policy $\{\pi_k\}_{k=1}^{K}$ has the weak regret and strong violation are bounded*

$$\mathcal{R}(K) := \sum_{k=1}^{K} \left(V_{r,1}^{\pi^*}(x_1^k) - V_{r,1}^{\pi_k}(x_1^k)\right) \leq R_K,$$

$$\mathcal{V}_+(K) := \sum_{k=1}^{K} \left(b - V_{g,1}^{\pi_k}(x_1^k)\right)_+ \leq B_K.$$

*Then the strong regret also satisfies*

$$\mathcal{R}_+(K) := \sum_{k=1}^{K} \left(V_{r,1}^{\pi^*}(x_1^k) - V_{r,1}^{\pi_k}(x_1^k)\right)_+ \leq R_K + \frac{H}{\rho} B_K.$$

The lemma shows that the strong regret is upper-bounded by the weak regret and the strong constraint violation, providing a clear trade-off between reward and utility/cost, with the constraint violation counted toward the regret. We provide a complete proof in Appendix D. To understand it, we consider a special example with $K = 4$ episodes and a single step $H = 1$ (i.e., bandit setting). The threshold $b = 1$ and the feasibility gap is $\rho = 1$. Consider an optimal policy such that $V_r^*(x_1) = V_g^*(x_1) = 1$ and the value functions under a policy $\{\pi_k\}$ are $V_r^{\pi_k}(x_1) = \{2, 2, 0, 0\}$ and $V_g^{\pi_k}(x_1) = \{0, 0, 2, 2\}$. Therefore, we have $\mathcal{R}(K) = 0$, $\mathcal{R}_+(K) = 2$ and $\mathcal{V}_+(K) = 2$, where the lemma holds as well. Intuitively, for these episodes where rewards acquired by a policy are even larger than the optimal policy, it must overuse the resource and result in constraint violation, which will count towards the strong regret.

Now with Lemma 5.4, we can establish strong regret and constraint violation under APMPO as follows.

**Theorem 5.5.** *Given any $p \in (0, 1)$, under the Slater's condition in Assumption 5.1, APMPO achieves the following strong regret and strong constraint violation with an unknown Slater's constant $\rho$ with a probability at least $1 - p$:*

$$\mathcal{R}_+(K) = \tilde{O}(\frac{\sqrt{KH^7}}{\rho^2} + \frac{\sqrt{d^3 H^5 K}}{\rho}),$$

$$\mathcal{V}_+(K) = \tilde{O}(\frac{\sqrt{KH^5}}{\rho} + \sqrt{d^3 H^3 K}).$$

*If the Slater's constant $\rho$ is known there is a tighter strong regret and strong constraint violation bound with a probability at least $1 - p$:*

$$\mathcal{R}_+(K) = \tilde{O}(\frac{\sqrt{KH^7}}{\rho} + \frac{\sqrt{d^3 H^5 K}}{\rho}),$$

$$\mathcal{V}_+(K) = \tilde{O}(\sqrt{KH^5} + \sqrt{d^3 H^3 K}).$$

*by adjusting the hyperparameter $\lambda$ according to $\rho$.*

*Remark* 5.6. Our near-optimality claim focuses on the dependence on the number of episodes $K$. The polynomial dependence on the horizon $H$ and feature dimension $d$ is likely not tight. It mainly comes from the standard confidence-radius analysis for linear MDPs and additional factors introduced by strong metrics, especially the weak-to-strong conversion in Lemma 5.4. Improving the $H$- and $d$-dependence is left as future work.

## 5.2. Proof Outline

In this section, we provide a proof sketch of the most important result in Theorem 5.2, which serves as the key cornerstone for both Theorem 5.5 and the subsequent results beyond Slater's condition. More details of the proof can be found in Appendix C.

To proceed in a concise form, we define the augmented (optimistic) $Q$-value function

$$\hat{Q}_h^k(x, a, \eta) = \hat{Q}_{r,h}^k(x, a) + \eta \hat{Q}_{g,h}^k(x, a),$$
$$Q_h^\pi(x, a, \eta) = Q_{r,h}^\pi(x, a) + \eta Q_{g,h}^\pi(x, a).$$

We denote the notation

$$\hat{Q}_h^k(x, \pi) = \sum_{a \in \mathcal{A}} \pi_h^k(a \mid x) \hat{Q}_h^k(x, a),$$
$$Q_h^\pi(x, \pi) = \sum_{a \in \mathcal{A}} \pi_h(a \mid x) Q_h^\pi(x, a)$$

for simplicity, which are also applied for $\hat{Q}_h^k(x, \pi_h, \eta_k)$ and $Q_h^\pi(x, \pi, \eta_k)$.

**No-regret property of APMPO.** To prove our main results, we first introduce the no-regret property of APMPO in the following lemma (proved in Appendix A), which provides an upper bound for the augmented optimistic $Q$-value function under APMPO against that under any policy (including the optimal policy $\pi^*$).

**Lemma 5.7.** *Let $\{\pi_{k,h}\}_{h=1}^H$ be the policy at episode $k$ generated by APMPO. Then under any policy $\pi = \{\pi_h\}_{h=1}^H$, the following inequality holds for any state $x$ and any step $h$*

$$\sum_{k=1}^K \hat{Q}_h^k(x, \pi_h, \eta_k) - \hat{Q}_h^k(x, \pi_{k,h}, \eta_k) \leq C_K, \quad (2)$$

*where $C_K = H(1 + \log|\mathcal{A}|)(\sqrt{K} + \sqrt{\sum_{k=1}^K \eta_k^2})$.*

The above lemma quantifies the performance difference of the augmented optimistic value function between APMPO and any comparator policy $\pi$, where the upper bound can be interpreted as a variance-aware violation bound by recalling $\eta_k = \lambda (b - \hat{V}_{g,1}^k(x_1^k))_+$. Intuitively, when APMPO is close to the optimal, we would expect an ideal bound of constraint violation (i.e., $\eta_k \approx 0$) and Lemma 5.7 also implies

$\sum_{k=1}^K \hat{Q}_{r,h}^k(x, \pi_h) - \hat{Q}_h^k(x, \pi_{k,h}) = O(\sqrt{K})$. This immediately suggests the regret performance bound if we associate the optimistic version of $\hat{Q}_{r,h}^k(x, \pi_h)$ and $\hat{Q}_{r,h}^k(x, \pi_{k,h})$ with its true value functions $Q_{r,h}(x, \pi_h)$ and $Q_{r,h}(x, \pi_{k,h})$. However, we have two major challenges to justify these intuitions: 1) the constraint violation process (i.e., $\{\eta_k\}$) is highly coupled with the reward process and environment learning; 2) it is unclear how to connect $\hat{Q}_h^k(x, \pi_h, \eta_k)$ with the corresponding true value function $Q_h^{\pi_h}(x, \pi_h, \eta_k)$. These two challenges can be addressed by our design of violation adaptive penalty and learning rate. Specifically, the inequality (2) implies a self-bound property where the violation-related terms (e.g., $\eta_k$) show both sides of the inequality that can be "canceled" and lead to sublinear bounds on both strong regret and constraint violation.

As discussed, $Q_h^{\pi_h}(x, \pi_h, \eta_k) \leq \hat{Q}_h^k(x, \pi_h, \eta_k)$ does not hold under APMPO because APMPO is an adaptive mirror descent policy optimization rather than a greedy policy. Fortunately, we have the cumulative performance $\sum_{k=1}^K \hat{Q}_h^k(x, \pi_h, \eta_k) - \hat{Q}_h^k(x, \pi_{k,h}, \eta_k)$ holds for any step $h \in [H]$, where we combine Bellman equation and leverage the backward induction to associate $\hat{Q}_h^k(x, \pi_h, \eta_k)$ with $Q_h^{\pi_h}(x, \pi_h, \eta_k)$ in the following lemma, whose detailed proof can be found in Appendix B.

**Lemma 5.8.** *Given any $p \in (0, 1)$, let $\{\pi_{k,h}\}_{h=1}^H$ be the policy at episode $k$ generated by APMPO. Under any policy $\pi = \{\pi_h\}_{h=1}^H$, the following bound holds for any state $x$ and any step $h$ with a probability at least $1 - p$*

$$\sum_{k=1}^K Q_h^\pi(x, \pi_h, \eta_k) - \hat{Q}_h^k(x, \pi_{k,h}, \eta_k) \leq (H - h + 1) C_K.$$

Lemma 5.8 compares the true augmented value function under a policy with the optimistic augmented value function under APMPO. Let $h = 1$ in the key recursive relationship in Lemma 5.8 and recall the definition of value functions

$$Q_1^\pi(x_1, \pi_1, \eta_k) = V_{r,1}^\pi(x_1) + \eta_k V_{g,1}^\pi(x_1),$$
$$\hat{Q}_1^k(x_1, \pi_1^k, \eta_k) = \hat{V}_{r,1}^k(x_1) + \eta_k \hat{V}_{g,1}^k(x_1).$$

We can obtain that

$$\sum_{k=1}^K V_{r,1}^\pi(x_1^k) - \hat{V}_{r,1}^k(x_1^k) + \sum_{k=1}^K \eta_k(V_{g,1}^\pi(x_1^k) - b)$$
$$- \sum_{k=1}^K \eta_k(\hat{V}_{g,1}^k(x_1^k) - b) \leq H C_K. \quad (3)$$

Recall the adaptive penalty factor $\eta_k = \lambda(b - \hat{V}_{g,1}^k(x_1^k))_+$. We have the key equality

$$\eta_k(b - \hat{V}_{g,1}^k(x_1^k)) = \lambda(b - \hat{V}_{g,1}^k(x_1^k))_+^2 = \frac{\eta_k^2}{\lambda}. \quad (4)$$

**Self-bound property.** We then plug the equality (4) into (3) and recall the value of $C_K$ such that

$$\sum_{k=1}^{K} V_{r,1}^{\pi}(x_1^k) - \hat{V}_{r,1}^k(x_1^k) + \sum_{k=1}^{K} \eta_k(V_{g,1}^{\pi}(x_1^k) - b)$$

$$+ \frac{1}{\lambda} \sum_{k=1}^{K} \eta_k^2 \leq H^2(1 + \log|\mathcal{A}|)(\sqrt{K} + \sqrt{\sum_{k=1}^{K} \eta_k^2}).$$

Now it is clear to see "self-bound property", where the terms of variance-aware violation appear on both sides. We add the term of $\lambda H^4(1 + \log|\mathcal{A}|)^2/4$ above and dropping the positive terms such that

$$\sum_{k=1}^{K} V_{r,1}^{\pi}(x_1^k) - \hat{V}_{r,1}^k(x_1^k) + \sum_{k=1}^{K} \eta_k(V_{g,1}^{\pi}(x_1^k) - b)$$

$$\leq 2H^2\sqrt{K}\log|\mathcal{A}| + \lambda H^4\log^2|\mathcal{A}|. \quad (5)$$

Next, we proceed to prove the weak regret and strong constraint violation.

**Regret analysis.** Let $\pi = \pi^*$ be the optimal policy in (5) and we know $V_{g,1}^{\pi^*}(x_1^k) \geq b$. Thus it implies

$$\sum_{k=1}^{K} V_{r,1}^{\pi^*}(x_1^k) - \hat{V}_{r,1}^k(x_1^k) \leq 2H^2\log|\mathcal{A}|\sqrt{K} + \lambda H^4\log^2|\mathcal{A}|.$$

Finally, we study the weak regret

$$\sum_{k=1}^{K} V_{r,1}^{\pi^*}(x_1^k) - V_{r,1}^{\pi_k}(x_1^k)$$

$$= \sum_{k=1}^{K} V_{r,1}^{\pi^*}(x_1^k) - \hat{V}_{r,1}^k(x_1^k) + \sum_{k=1}^{K} \hat{V}_{r,1}^k(x_1^k) - V_{r,1}^{\pi_k}(x_1^k)$$

$$\leq 2H^2\log|\mathcal{A}|\sqrt{K} + \lambda H^4\log^2|\mathcal{A}| + \tilde{O}(\sqrt{d^3H^3K})$$

$$\leq \tilde{O}(\sqrt{KH^5}) + \tilde{O}(\sqrt{d^3H^3K}),$$

where the first inequality follows from lemma 5.9 regarding optimistic linear estimation error (proved in Appendix F), and the second inequality is obtained by setting $\lambda = \sqrt{KH^{-3}}$. Combining these bounds, the weak regret violation can be bounded by $\tilde{O}(\sqrt{K})$. Furthermore, if the Slater's constant $\rho$ is known, we obtain another bound of $\tilde{O}(\frac{\sqrt{K}}{\rho})$ with $\lambda = \sqrt{KH^{-3}\rho^{-2}}$.

**Lemma 5.9.** *Given any $p \in (0,1)$, with probability at least $1 - p$, the cumulative errors between the optimistic value functions and the values induced by* APMPO *satisfy*

$$\sum_{k=1}^{K} \left( \hat{V}_{u,1}^k(x_1^k) - V_{u,1}^{\pi_k}(x_1^k) \right) = \tilde{O}\left( \sqrt{d^3H^3K} \right),$$

*where $u \in \{r, g\}$.*

**Violation analysis.** Let $\pi$ be a strictly feasible policy satisfying Slater's condition with $V_{g,1}^{\pi}(x_1^k) \geq b + \rho$. Thus the inequality (5) implies

$$\sum_{k=1}^{K} V_{r,1}^{\pi}(x_1^k) - \hat{V}_{r,1}^k(x_1^k) + \rho \sum_{k=1}^{K} \eta_k$$

$$\leq 2H^2\sqrt{K}\log|\mathcal{A}| + \lambda H^4\log^2|\mathcal{A}|.$$

Recall the definition of $\eta_k$ and the value functions are bounded such that $|\sum_{k=1}^{K} V_{r,1}^{\pi}(x_1^k) - \hat{V}_{r,1}^k(x_1^k)| \leq HK$. We have the (optimistic) strong constraint violation

$$\sum_{k=1}^{K} (b - \hat{V}_{g,1}^k(x_1^k))_+ = \tilde{O}(\sqrt{d^3H^3K}). \quad (6)$$

Finally, we establish a bound for the strong violation:

$$\sum_{k=1}^{K} (b - V_{g,1}^{\pi_k}(x_1^k))_+$$

$$\leq \sum_{k=1}^{K} (b - \hat{V}_{g,1}^k(x_1^k))_+ + \sum_{k=1}^{K} (\hat{V}_{g,1}^k(x_1^k) - V_{g,1}^{\pi_k}(x_1^k))_+$$

$$\leq \frac{HK + 2H^2\sqrt{K}\log|\mathcal{A}| + \lambda H^4\log^2|\mathcal{A}|}{\lambda\rho} + \tilde{O}(\sqrt{d^3H^3K})$$

$$\leq \tilde{O}(\frac{\sqrt{KH^5}}{\rho}) + \tilde{O}(\sqrt{d^3H^3K}),$$

where the first inequality follows from lemma 5.9, and the second inequality is obtained by setting $\lambda = \sqrt{KH^{-3}}$. Combining these bounds, the strong constraint violation can be bounded by $\tilde{O}(\frac{\sqrt{K}}{\rho})$. Furthermore, if the Slater's constant $\rho$ is known, a tighter bound of $\tilde{O}(\sqrt{K})$ with $\lambda = \sqrt{KH^{-3}\rho^{-2}}$.

Under the obtained bounds for weak regret and strong constraint violation, we can now derive the strong regret guarantee stated in Theorem 5.5 via Lemma 5.4.

### 5.3. Results Beyond Slater's Condition

The results above are based on Slater's condition in analyzing the strong constraint violation and translating the weak regret into the strong one with the strong duality property. We further proceed to address the original CMDP problem in (1) without Slater's condition.

**CMDP beyond Slater's Condition.** We consider a $\delta$-slack version of the original CMDP with the threshold $b_\delta = b - \delta$ in the constraint

$$\max_{\pi} \quad V_{r,1}^{\pi}(x_1), \quad \text{s.t.} \quad V_{g,1}^{\pi}(x_1) \geq b_\delta. \quad (7)$$

Intuitively, the $\delta$-slackness problem in (7) does satisfy Slater's condition, though the original problem in (1) does not, as justified in the following lemma.

**Lemma 5.10.** *Suppose the original CMDP in* (1) *is feasible, i.e., there exists a policy $\pi'$ such that $V_{g,1}^{\pi'}(x_1) \geq b$. Then for any $\delta > 0$, the $\delta$-slack CMDP in* (7) *satisfies Slater's condition. In particular, the same policy $\pi'$ is strictly feasible for the $\delta$-slack CMDP with Slater constant $\rho = \delta$, since*

$$V_{g,1}^{\pi'}(x_1) \geq b = b_\delta + \delta.$$

This lemma is straightforward to verify by just letting $\pi'$ be any feasible policy of the original CMDP in (1), where it satisfies $V_{g,1}^{\pi'}(x_1) \geq b \geq b_\delta + \rho$.

Our algorithm requires a slight twist on the (slack) penalty factor $\eta_k = \lambda(b_\delta - \hat{V}_{g,1}^k(x_1^k))_+$ in `APMPO`, which we call `APMPO`$(\lambda, \delta)$.

**Guarantees for the $\delta$-slack CMDP.** Consider the $\delta$-slack CMDP in (7). By employing $\delta$ as Slater's constant, the bound in Theorem 5.5 can be applied to algorithm `APMPO`$(\lambda, \delta)$, yielding bounds for strong regret and strong constraint violation in this setting. Let $\pi_\delta^*$ denote the optimal policy of the $\delta$-slack CMDP in (7) and let $\pi^k$ be the policy deployed by `APMPO`$(\lambda, \delta)$ in episode $k$. We then have

$$\mathcal{R}_+^{\delta\text{-slack}}(K) = \sum_{k=1}^{K} \left( V_{r,1}^{\pi_\delta^*}(x_1^k) - V_{r,1}^{\pi^k}(x_1^k) \right)_+$$
$$\leq \tilde{O}\left( \frac{H^2 K + 2H^3\sqrt{K} + \lambda H^5}{\lambda \delta^2} + \frac{\sqrt{d^3 H^5 K}}{\delta} \right),$$

$$\mathcal{V}_+^{\delta\text{-slack}}(K) = \sum_{k=1}^{K} \left( b_\delta - V_{g,1}^{\pi^k}(x_1^k) \right)_+$$
$$\leq \tilde{O}\left( \frac{HK + 2H^2\sqrt{K} + \lambda H^4}{\lambda \delta} + \sqrt{d^3 H^3 K} \right).$$

**Guarantees Without Slater's Condition.** The optimal policy $\pi^*$ of the original CMDP (1) is also feasible for the $\delta$-slack CMDP (7), implying $V_{r,1}^{\pi^*}(x_1^k) \leq V_{r,1}^{\pi_\delta^*}(x_1^k)$ for all $k$. Additionally, recall the definition of $b_\delta = b - \delta$. Therefore, the strong regret and constraint violation of the original CMDP (1) satisfies Theorem 5.11, and we proved it in Appendix E.

**Theorem 5.11.** *Given any $p \in (0,1)$, with $\lambda = K^{\frac{3}{4}} H^{-1}$, $\delta = K^{-\frac{1}{4}}$, `APMPO`$(\lambda, \delta)$ achieves the following strong regret and constraint violation with a probability at least $1 - p$:*

$$\mathcal{R}_+(K) = \tilde{O}(K^{\frac{3}{4}}), \quad \mathcal{V}_+(K) = \tilde{O}(K^{\frac{3}{4}}).$$

## 6. Numerical Simulation

This section aims to justify the effectiveness of our proposed `APMPO` and theoretical performance through numerical experiments. We compare `APMPO` against the previous state-of-the-art linear CMDP algorithms by Ghosh et al. (2024)

and Stradi et al. (2025a). Further experimental details can be found in Appendix G, and the experimental results are presented in Figure 1. From the results, we observe that `APMPO` achieves sublinear strong regret bounds and violation bounds in all environments, which aligns with our theoretical conclusions. Furthermore, compared to other baseline algorithms, our proposed `APMPO` achieves a better trade-off, characterized by lower regret and violation.

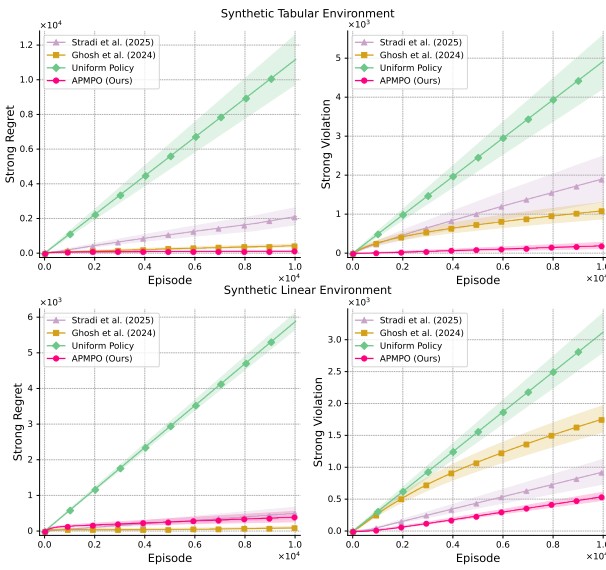

*Figure 1.* Numerical simulation results in the synthetic tabular environment (Top) and the synthetic linear environment (Bottom).

## 7. Limitations and Future Work

Although `APMPO` achieves sharp dependence on $K$, its dependence on $H$ and $d$ is not optimized. The current $O(d^{3/2})$ factor may be reducible via Bernstein-type or variance-aware confidence bounds, potentially toward the $O(d)$ lower bound suggested by bandit/linear MDP settings (He et al., 2021). For $H$, strong metrics penalize per-episode errors without cancellation and introduce extra factors in the weak-to-strong conversion; sharp $H, d$ lower bounds remain open. Another limitation is the Slater-free rate: our $\tilde{O}(K^{3/4})$ bound is the first sublinear strong-metric guarantee in this setting, but it remains open whether the near-optimal $\tilde{O}(\sqrt{K})$ dependence under Slater's condition is attainable without a feasibility gap.

## 8. Conclusion

We study episodic linear CMDPs under strong regret and constraint violation. Our model-free algorithm `APMPO` achieves near-optimal $\tilde{O}(\sqrt{K})$ bounds under Slater's condition and the first sublinear $\tilde{O}(K^{3/4})$ bounds without it, using a violation-aware penalty design.

## Acknowledgment

This work was supported by the National Natural Science Foundation of China under Grant 62302305, Key Laboratory of Interdisciplinary Research of Computation and Economics (Shanghai University of Finance and Economics), Ministry of Education, HPC Platform of ShanghaiTech University, ShanghaiTech University GenAI Platform.

## Impact Statement

This paper presents work whose goal is to advance the field of Reinforcement Learning. There are many potential societal consequences of our work, none of which we feel must be specifically highlighted here.

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

## A. Proof of Lemma 5.7: no-regret property of `APMPO`

In this section, we provide detailed proof of Lemma 5.7, which provides a unified bound of penalty augmented optimistic value function of `APMPO` against that under the optimal poicy. We begin by stating the following three-point (please refer to Xiao (2022) for a more detailed discussion and analysis).

**Lemma A.1.** *Let $\mathcal{X}$ be a convex set. Let function $h$ be convex on $\Pi$, and let $\pi_{opt} \in \Pi$ be a global minimum of $h(\pi) + D(\pi||\pi_{t-1})$ on $\Pi$. Then, for any $\pi \in \Pi$, we have:*

$$h(\pi_{opt}) + \alpha D(\pi_{opt}||\pi_{t-1}) \leq h(\pi) + \alpha D(\pi||\pi_{t-1}) - \alpha D(\pi||\pi_{opt})$$

*Proof.* Due to the definition of $\pi_{opt}$, we have

$$\nabla h(\pi_{opt}) + \alpha \nabla D(\pi_{opt}||\pi_{t-1}) = 0,$$

A standard "three-point" expansion of the second bracket shows

$$D(\pi||\pi_{t-1}) - D(\pi_{opt}||\pi_{t-1}) = D(\pi||\pi_{opt}) + \langle \nabla D(\pi_{opt}||\pi_{t-1}), \pi - \pi_{opt} \rangle,$$

Recall $\nabla h(\pi_{opt}) + \nabla D(\pi_{opt}||\pi_{t-1}) = 0$, then we can get

$$\begin{aligned}
&(h(\pi) - h(\pi_{opt})) + \alpha(D(\pi||\pi_{t-1}) - D(\pi_{opt}||\pi_{t-1})) \\
&= (h(\pi) - h(\pi_{opt}) - \langle \nabla h(\pi_{opt}), \pi - \pi_{opt} \rangle) + \alpha D(\pi||\pi_{opt}) \\
&\geq \alpha D(\pi||\pi_{opt}),
\end{aligned}$$

Rearranging the above inequality, we complete the proof. $\square$

Recall for convenience, we define the (value) function $\Phi(x, a)$ under the policy expectation under policy $\pi$ as

$$\Phi(x, \pi) := \sum_{a \in \mathcal{A}} \pi(a|x)\, \phi(x, a). \tag{8}$$

We now prove a no-regret property of `APMPO` for optimistic value functions for any step $h$.

**Lemma A.2.** *(Restate of Lemma 5.7) Let $\{\pi_{k,h}\}_{h=1}^{H}$ be the policy at episode $k$ generated by `APMPO`. Consider any comparator policy $\pi = \{\pi_h\}_{h=1}^{H}$. Then for any state $x$ and any step $h \in [H]$, the following inequality holds:*

$$\sum_{k=1}^{K} \hat{Q}_{r,h}^{k}(x, \pi_h) + \eta_k \hat{Q}_{g,h}^{k}(x, \pi_h) - \hat{Q}_{r,h}^{k}(x, \pi_{k,h}) - \eta_k \hat{Q}_{g,h}^{k}(x, \pi_{k,h})$$

$$\leq (1 + \log|\mathcal{A}|)H \sqrt{K + \sum_{k=1}^{K} \eta_k^2}. \tag{9}$$

*Proof.* For each episode $k$, define the (negative) penalty augmented optimistic value function

$$\ell_h^k(x, a) = -\hat{Q}_{r,h}^k(x, a) - \eta_k \hat{Q}_{g,h}^k(x, a), \qquad \forall x \in \mathcal{S}, a \in \mathcal{A}. \tag{10}$$

The policy update of `APMPO` at step $h$ is given by the KL-regularized mirror descent optimization

$$\pi_{k+1,h}(\cdot|x) = \underset{\pi_h(\cdot|x) \in \Delta(\mathcal{A})}{\arg\min} \left\{ \langle \ell_h^k(x, \cdot), \pi_h(\cdot|x) - \pi_{k,h}(\cdot|x) \rangle + \alpha_k \mathrm{KL}(\pi_h(\cdot|x) \,\|\, \pi_{k,h}(\cdot|x)) \right\}. \tag{11}$$

By three-point descent Lemma A.1, for any comparator policy $\pi_h(\cdot|x)$ (including the optimal policy $\pi^*$), we have

$$\begin{aligned}
&\langle \ell_h^k(x, \cdot), \pi_{k+1,h}(\cdot|x) - \pi_{k,h}(\cdot|x) \rangle + \alpha_k \mathrm{KL}(\pi_{k+1,h}(\cdot|x) \,\|\, \pi_{k,h}(\cdot|x)) \\
&\leq \langle \ell_h^k(x, \cdot), \pi_h(\cdot|x) - \pi_{k,h}(\cdot|x) \rangle + \alpha_k \mathrm{KL}(\pi_h(\cdot|x) \,\|\, \pi_{k,h}(\cdot|x)) - \alpha_k \mathrm{KL}(\pi_h(\cdot|x) \,\|\, \pi_{k+1,h}(\cdot|x)).
\end{aligned} \tag{12}$$

We now bound the stability term appearing in the three-point inequality in (12). For any step $h$ and state $x$ at episode $k$, consider the quantity

$$\langle \ell_h^k(x,\cdot), \pi_{k+1,h}(\cdot|x) - \pi_{k,h}(\cdot|x)\rangle + \alpha_k \mathrm{KL}(\pi_{k+1,h}(\cdot|x) \,\|\, \pi_{k,h}(\cdot|x)). \tag{13}$$

By Fenchel–Young inequality for the negative entropy mirror map, for any probability distributions $p, q \in \Delta(\mathcal{A})$ and any vector $v \in \mathbb{R}^{|\mathcal{A}|}$, we have

$$\langle v, p - q\rangle \leq \frac{1}{2\alpha_k}\|v\|_\infty^2 + \alpha_k \mathrm{KL}(p\|q). \tag{14}$$

Applying (14) with $p = \pi_{k+1,h}(\cdot|x)$, $q = \pi_{k,h}(\cdot|x)$, and $v = -\ell_h^k(x,\cdot)$ yields

$$\langle -\ell_h^k(x,\cdot), \pi_{k+1,h}(\cdot|x) - \pi_{k,h}(\cdot|x)\rangle \leq \frac{\|\ell_h^k(x,\cdot)\|_\infty^2}{2\alpha_k} + \alpha_k \mathrm{KL}(\pi_{k+1,h}(\cdot|x) \,\|\, \pi_{k,h}(\cdot|x)). \tag{15}$$

Rearranging (15) yields

$$\langle -\ell_h^k(x,\cdot), \pi_{k+1,h}(\cdot|x) - \pi_{k,h}(\cdot|x)\rangle - \alpha_k \mathrm{KL}(\pi_{k+1,h}(\cdot|x) \,\|\, \pi_{k,h}(\cdot|x)) \leq \frac{\|\ell_h^k(x,\cdot)\|_\infty^2}{2\alpha_k}. \tag{16}$$

Therefore, adding this inequality to inequality (12), such that the following inequality holds

$$\langle \ell_h^k(x,\cdot), \pi_{k,h}(\cdot|x) - \pi_h(\cdot|x)\rangle \leq \frac{\|\ell_h^k(x,\cdot)\|_\infty^2}{2\alpha_k} + \alpha_k \mathrm{KL}(\pi_h(\cdot|x) \,\|\, \pi_{k,h}(\cdot|x)) - \alpha_k \mathrm{KL}(\pi_h(\cdot|x) \,\|\, \pi_{k+1,h}(\cdot|x)) \tag{17}$$

Then, summing (17) over $k = 1, \dots, K$ and telescoping the KL terms yields the weak regret bound for KL-regularized online mirror descent

$$\sum_{k=1}^K \langle \ell_h^k(x,\cdot), \pi_{k,h}(\cdot|x) - \pi_h(\cdot|x)\rangle \tag{18}$$

$$\leq \sum_{k=1}^K \alpha_k \mathrm{KL}(\pi_h(\cdot|x) \,\|\, \pi_{k,h}(\cdot|x)) - \alpha_k \mathrm{KL}(\pi_h(\cdot|x) \,\|\, \pi_{k+1,h}(\cdot|x)) + \sum_{k=1}^K \frac{\|\ell_h^k(x,\cdot)\|_\infty^2}{2\alpha_k}. \tag{19}$$

Since $\{\alpha_k\}$ with $\alpha_k = \sqrt{\sum_{s=1}^k (1 + \eta_s^2)}$ is a non-decreasing sequence and KL divergence is non-negative, we have $0 \leq \mathrm{KL}(\pi_h(\cdot|x)\|\pi_{k,h}(\cdot|x)) \leq \log|\mathcal{A}|$ for all $h \in [H]$. This implies

$$\alpha_1 \mathrm{KL}(\pi_h(\cdot|x)\|\pi_{1,h}(\cdot|x)) + \sum_{k=2}^K (\alpha_k - \alpha_{k-1})\mathrm{KL}(\pi_h(\cdot|x)\|\pi_{k,h}(\cdot|x)) \leq \alpha_K \log|\mathcal{A}|. \tag{20}$$

Finally, we have

$$\sum_{k=1}^K \langle \ell_h^k(x,\cdot), \pi_{k,h}(\cdot|x) - \pi_h(\cdot|x)\rangle \leq \alpha_K \log|\mathcal{A}| + \sum_{k=1}^K \frac{\|\ell_h^k(x,\cdot)\|_\infty^2}{2\alpha_k}. \tag{21}$$

Recall the notation of policy expectation in (8). By the definition of augmented optimistic value function in (10) and linearity of expectation, we have for any state $x$ and step $h$

$$\langle \ell_h^k(x,\cdot), \pi_{k,h}(\cdot|x) - \pi_h(\cdot|x)\rangle \tag{22}$$

$$= \hat{Q}_{r,h}^k(x,\pi_h) + \eta_k \hat{Q}_{g,h}^k(x,\pi_h) - \hat{Q}_{r,h}^k(x,\pi_{k,h}) - \eta_k \hat{Q}_{g,h}^k(x,\pi_{k,h}). \tag{23}$$

Substituting (23) into (21) gives

$$\sum_{k=1}^{K} \hat{Q}_{r,h}^k(x_h^k, \pi_h) + \eta_k \hat{Q}_{g,h}^k(x_h^k, \pi_h) - \hat{Q}_{r,h}^k(x_h^k, \pi_{k,h}) - \eta_k \hat{Q}_{g,h}^k(x_h^k, \pi_{k,h})$$

$$\leq \alpha_K \log |\mathcal{A}| + H^2 \sum_{k=1}^{K} \frac{1+\eta_k^2}{2\alpha_k}. \tag{24}$$

where we use $|\hat{Q}_{r,h}^k(x,a)| \leq H$ and $|\hat{Q}_{g,h}^k(x,a)| \leq H$ such that $\|\ell_h^k(x, \cdot)\|_\infty^2 \leq H^2(1+\eta_k)^2$ for any $k, h, x, a$. Recall the choice of $\alpha_k = \sqrt{k + \sum_{s=1}^{k} \eta_s^2}$ and by Lemma A.3, we have proved the lemma. $\qquad\square$

The following lemma is a classical result in the adaptive gradient descent (Orabona, 2019), we prove it for completeness.

**Lemma A.3.** *Let* $\alpha_k = \sqrt{k + \sum_{i=1}^{k} \eta_i^2}$. *Then, we have*

$$\sum_{k=1}^{K} \frac{1+\eta_k^2}{\alpha_k} \leq 2\sqrt{K + \sum_{i=1}^{K} \eta_i^2}.$$

*Proof.* Define

$$S_k := k + \sum_{i=1}^{k} \eta_i^2, \text{ and } \alpha_k = \sqrt{S_k}.$$

This implies $S_k - S_{k-1} = 1 + \eta_k^2$, and we have

$$\sqrt{S_k} - \sqrt{S_{k-1}} = \frac{S_k - S_{k-1}}{\sqrt{S_k} + \sqrt{S_{k-1}}} = \frac{1+\eta_k^2}{\sqrt{S_k} + \sqrt{S_{k-1}}} \geq \frac{1+\eta_k^2}{2\sqrt{S_k}}.$$

Therefore, we have

$$\frac{1+\eta_k^2}{\sqrt{S_k}} := \frac{1+\eta_k^2}{\alpha_k} \leq 2\big(\sqrt{S_k} - \sqrt{S_{k-1}}\big).$$

Summing over $k = 1, \ldots, K$ yields

$$\sum_{k=1}^{K} \frac{1+\eta_k^2}{\alpha_k} \leq 2 \sum_{k=1}^{K} \big(\sqrt{S_k} - \sqrt{S_{k-1}}\big) = 2\sqrt{S_K},$$

which completes the proof. $\qquad\square$

# B. Proof of Lemma 5.8

In this section, we connect the optimistic augmented value functions with its true version. Define the augmented value functions

$$V_h^\pi(x, \eta) = V_{r,h}^\pi(x) + \eta V_{g,h}^\pi(x), \ \ V_h^k(x, \eta) = V_{r,h}^k(x) + \eta V_{g,h}^k(x), \tag{25}$$

$$Q_h^\pi(x, a, \eta) = Q_{r,h}^\pi(x, a) + \eta Q_{g,h}^\pi(x, a), \ \ Q_h^\pi(x, a, \eta) = Q_{r,h}^\pi(x, a) + \eta Q_{g,h}^\pi(x, a). \tag{26}$$

Similarly, for the estimated parameters, due to the linear assumption and Lemma F.1, the augmented weight $\mathbf{w}_h^k(\eta)$ satisfies

$$\mathbf{w}_h^k(\eta) = \mathbf{w}_{r,h}^k + \eta \mathbf{w}_{g,h}^k. \tag{27}$$

**Lemma B.1.** *For any policy* $\pi$, *at step* $h$ *of episode* $k$, *and parameter* $\eta \geq 0$. *On the event* $\mathcal{E}$, *for any state–action pair* $(x, a)$, *the following inequality holds:*

$$Q_h^\pi(x, a, \eta) \leq \hat{Q}_h^k(x, a, \eta) + \mathbb{P}_h\big(V_{h+1}^\pi - V_{h+1}^k\big)(x, a). \tag{28}$$

*Proof.* We first bound the part of reward value functions and that of utility value functions follows. By Lemma F.3, we have

$$\langle \phi(x,a), \mathbf{w}_h^k \rangle - Q_h^\pi(x,a) = \mathbb{P}_h(V_{h+1}^k - V_{h+1}^\pi)(x,a) + \Delta_h^k(x,a), \tag{29}$$

for some $\Delta_h^k(x,a)$ that satisfies $|\Delta_h^k(x,a)| \le \beta_k \sqrt{\phi(x,a)^\top (\Lambda_h^k)^{-1} \phi(x,a)}$. We have

$$\langle \phi(x,a), \mathbf{w}_h^k \rangle - \Delta_h^k(x,a) \le \hat{Q}_{r,h}^k(x,a) \tag{30}$$

holds with a high probability according to the definition of $\hat{Q}_{r,h}^k(x,a)$. Therefore, we have

$$Q_{r,h}^\pi(x,a) \le \hat{Q}_{r,h}^k(x,a) + \mathbb{P}_h \left( V_{r,h+1}^\pi - V_{r,h+1}^k \right)(x,a). \tag{31}$$

Similarly, applying the same steps to the utility value function yields

$$Q_{g,h}^\pi(x,a) \le \hat{Q}_{g,h}^k(x,a) + \mathbb{P}_h \left( V_{g,h+1}^\pi - V_{g,h+1}^k \right)(x,a). \tag{32}$$

Multiplying (32) by $\eta$ and summing it with (31), we obtain

$$Q_{r,h}^\pi(x,a) + \eta Q_{g,h}^\pi(x,a) \le \hat{Q}_{r,h}^k(x,a) + \eta \hat{Q}_{g,h}^k(x,a) \tag{33}$$
$$+ \mathbb{P}_h \left( V_{r,h+1}^\pi - V_{r,h+1}^k \right)(x,a) + \eta \mathbb{P}_h \left( V_{g,h+1}^\pi - V_{g,h+1}^k \right)(x,a). \tag{34}$$

By the definition of the augmented value functions,

$$V_{h+1}^\pi = V_{r,h+1}^\pi + \eta V_{g,h+1}^\pi, \quad V_{h+1}^k = V_{r,h+1}^k + \eta V_{g,h+1}^k, \tag{35}$$

the transition terms combine as

$$\mathbb{P}_h \left( V_{r,h+1}^\pi - V_{r,h+1}^k \right) + \eta \mathbb{P}_h \left( V_{g,h+1}^\pi - V_{g,h+1}^k \right) = \mathbb{P}_h \left( V_{h+1}^\pi - V_{h+1}^k \right). \tag{36}$$

Substituting this identity completes the proof of (39). □

**Lemma B.2.** *Let $C_K := (1 + \log |\mathcal{A}|) H \sqrt{K + \sum_{k=1}^K \eta_k^2}$. Let $\{\pi_{k,h}\}_{h=1}^H$ be the policy at episode $k$ generated by* APMPO. *Consider any comparator policy $\pi = \{\pi_h\}_{h=1}^H$. For any step $h \in [H]$, the following bound holds:*

$$\sum_{k=1}^K Q_h^\pi(x, \pi_h, \eta_k) - \hat{Q}_h^k(x, \pi_{k,h}, \eta_k) \le (H - h + 1) C_K. \tag{37}$$

*Consequently,*

$$\sum_{k=1}^K V_1^\pi(x_1^k, \eta_k) - \hat{V}_1^k(x_1^k, \eta_k) \le H C_K. \tag{38}$$

*Proof.* We prove (37) by backward induction on $h$. From Lemma B.1, we have for any step $h$

$$\sum_{k=1}^K Q_h^\pi(x, a, \eta) - \hat{Q}_h^k(x, a, \eta) \le \mathbb{P}_h \left( V_{h+1}^\pi - V_{h+1}^k \right)(x,a). \tag{39}$$

For the base case $h = H$, since $V_{H+1}^\pi = V_{H+1}^k = 0$, the inequality (39) reduces to

$$\sum_{k=1}^K Q_H^\pi(x, \pi_H, \eta_k) - \hat{Q}_H^k(x, \pi_{k,h}, \eta_k) \le 0 \le C_K.$$

Assume the inequality (37) holds at step $h$

$$\sum_{k=1}^{K} Q_h^{\pi}(x, \pi_h, \eta_k) - \hat{Q}_h^k(x, \pi_{k,h}, \eta_k) \le (H - h + 1)C_K. \tag{40}$$

Now we study the case at $h - 1$ step. Again by (39), we have for any action $a$ at step $h - 1$

$$\sum_{k=1}^{K} Q_{h-1}^{\pi}(x, a, \eta) + \sum_{k=1}^{K} \mathbb{P}_h(V_h^k - V_h^{\pi})(x, a) \le \sum_{k=1}^{K} \hat{Q}_{h-1}^k(x, a, \eta), \tag{41}$$

which implies

$$\sum_{k=1}^{K} Q_{h-1}^{\pi}(x, \pi_{h-1}, \eta_k) + \sum_{k=1}^{K} \mathbb{P}_h(V_h^k - V_h^{\pi})(x, \pi_{h-1}) \le \sum_{k=1}^{K} \hat{Q}_{h-1}^k(x, \pi_{h-1}, \eta_k). \tag{42}$$

By the no-regret property of APMPO in Lemma 5.7, we have at step $h - 1$

$$\sum_{k=1}^{K} \hat{Q}_{h-1}^{\pi}(x, \pi_{h-1}, \eta_k) - \hat{Q}_{h-1}^k(x, \pi_{k,h-1}, \eta_k)$$
$$:= \sum_{k=1}^{K} \hat{Q}_{r,h-1}^k(x, \pi_{h-1}) + \eta_k \hat{Q}_{g,h-1}^k(x, \pi_{h-1}) - \hat{Q}_{r,h-1}^k(x, \pi_{k,h-1}) - \eta_k \hat{Q}_{g,h-1}^k(x, \pi_{k,h-1}) \le C_K \tag{43}$$

Combine (42) and (43) such that

$$\sum_{k=1}^{K} Q_{h-1}^{\pi}(x, \pi_{h-1}, \eta_k) - \hat{Q}_{h-1}^k(x, \pi_{k,h-1}, \eta_k) \le C_K + \sum_{k=1}^{K} \mathbb{P}_{h-1}(V_h^{\pi} - V_h^k)(x, \pi_{h-1}). \tag{44}$$

By the definition of the transition kernel operator, we have

$$\mathbb{P}_{h-1}(V_h^{\pi} - V_h^k)(x, a) = \sum_{x'} \mathbb{P}_{h-1}(x'|x, a) \left( Q_h^{\pi}(x', \pi_h, \eta_k) - Q_h^k(x', \pi_{k,h}, \eta_k) \right). \tag{45}$$

It implies

$$\sum_{k=1}^{K} \mathbb{P}_{h-1}(V_h^{\pi} - V_h^k)(x, a) = \sum_{x'} \mathbb{P}_{h-1}(x'|x, a) \sum_{k=1}^{K} \left( Q_h^{\pi}(x', \pi_h, \eta_k) - Q_h^k(x', \pi_{k,h}, \eta_k) \right). \tag{46}$$

Therefore, we have

$$\sum_{k=1}^{K} \mathbb{P}_{h-1}(V_h^{\pi} - V_h^k)(x, \pi_{h-1}) = \sum_{a} \pi_{h-1}(a|x) \sum_{x'} \mathbb{P}_{h-1}(x'|x, a) \sum_{k=1}^{K} \left( Q_h^{\pi}(x', \pi_h, \eta_k) - Q_h^k(x', \pi_{k,h}, \eta_k) \right). \tag{47}$$

Applying the induction hypothesis in (40) such that

$$\sum_{k=1}^{K} \mathbb{P}_{h-1}(V_h^{\pi} - V_h^k)(x, \pi_{h-1}) \le \sum_{a} \pi_{h-1}(a|x) \sum_{x'} \mathbb{P}_{h-1}(x'|x, a)(H - h + 1) \cdot C_K = (H - h + 1)C_K.$$

Substituting into (44) yields

$$\sum_{k=1}^{K} Q_{h-1}^{\pi}(x, \pi_{h-1}, \eta_k) - \hat{Q}_{h-1}^k(x, \pi_{h-1}^k, \eta_k) \le (H - h + 2) C_K,$$

which proves the step at $h - 1$ and completes the induction. Finally, let $h = 1$ and using the definition of value function proves (38). $\qquad \square$

# C. Proof of Theorem 5.2

**Theorem C.1** (Restate of Theorem 5.2). *Given any $p \in (0, 1)$,* APMPO *achieves the following weak regret and strong constraint violation with at least probability $1 - p$:*

$$\mathcal{R}(K) = \tilde{O}(\sqrt{KH^5} + \sqrt{d^3H^3K}), \quad \mathcal{V}_+(K) = \tilde{O}(\frac{\sqrt{KH^5}}{\rho} + \sqrt{d^3H^3K}).$$

*If the Slater's constant $\rho$ is known, there is a tighter strong regret and strong constraint violation bound with at least probability $1 - p$*

$$\mathcal{R}(K) = \tilde{O}(\frac{\sqrt{KH^5}}{\rho} + \sqrt{d^3H^3K}), \quad \mathcal{V}_+(K) = \tilde{O}(\sqrt{KH^5} + \sqrt{d^3H^3K}).$$

*Proof.* From the key recursive relationship of (38) in Lemma 5.8, we have

$$\sum_{k=1}^{K} V_{r,1}^{\pi}(x_1^k) + \eta_k(V_{g,1}^{\pi}(x_1^k) - b) - \hat{V}_1^k(x_1^k, \eta_k) - \eta_k(\hat{V}_{g,1}^k(x_1^k) - b) \leq H\,C_K, \tag{48}$$

where $C_K := (1 + \log|\mathcal{A}|)H\sqrt{K + \sum_{k=1}^{K}\eta_k^2}$. Recall the adaptive penalty factor $\eta_k = \lambda(b - \hat{V}_{g,1}^k(x_1^k))_+$, we have

$$\eta_k(b - \hat{V}_{g,1}^k(x_1^k)) = \lambda(b - \hat{V}_{g,1}^k(x_1^k))_+^2. \tag{49}$$

Now, we proceed to prove the weak regret and strong constraint violation.

**Regret analysis**: Let $\pi = \pi^*$ in (3) and we know $V_{g,1}^{\pi^*}(x_1^k) \geq b$. Thus it implies

$$\sum_{k=1}^{K} V_{r,1}^{\pi}(x_1^k) - \hat{V}_{r,1}^k(x_1^k) - \sum_{k=1}^{K} \eta_k(\hat{V}_{g,1}^k(x_1^k) - b) \leq H^2(1 + \log|\mathcal{A}|)(\sqrt{K} + \sqrt{\sum_{k=1}^{K}\eta_k^2}), \tag{50}$$

We then plug (4) such that

$$\sum_{k=1}^{K} V_{r,1}^{\pi}(x_1^k) - \hat{V}_{r,1}^k(x_1^k) + \lambda\sum_{k=1}^{K}(b - \hat{V}_{g,1}^k(x_1^k))_+^2 \tag{51}$$

$$\leq H^2(1 + \log|\mathcal{A}|)(\sqrt{K} + \lambda\sqrt{\sum_{k=1}^{K}(b - \hat{V}_{g,1}^k(x_1^k))_+^2}). \tag{52}$$

We add the term of $\lambda H^4(1 + \log|\mathcal{A}|)^2/4$ above and dropping the positive terms such that

$$\sum_{k=1}^{K} V_{r,1}^{\pi}(x_1^k) - \hat{V}_{r,1}^k(x_1^k) \leq H^2(1 + \log|\mathcal{A}|)\sqrt{K} + \frac{\lambda H^4(1 + \log|\mathcal{A}|)^2}{4},$$

which imples

$$\sum_{k=1}^{K} V_{r,1}^{\pi}(x_1^k) - V_{r,1}^{\pi_k}(x_1^k) = \sum_{k=1}^{K} V_{r,1}^{\pi}(x_1^k) - \hat{V}_{r,1}^k(x_1^k) + \sum_{k=1}^{K} \hat{V}_{r,1}^k(x_1^k) - V_{r,1}^{\pi_k}(x_1^k) \tag{53}$$

$$\leq H^2(1 + \log|\mathcal{A}|)\sqrt{K} + \frac{\lambda H^4(1 + \log|\mathcal{A}|)^2}{4} + \tilde{O}(\sqrt{d^3H^3K}) \tag{54}$$

This completes the regret analysis by invoking Lemma F.5 on the optimistic learning errors $\tilde{O}(\sqrt{d^3H^3K})$.

If the Slater's constant $\rho$ is unknown, setting $\lambda = \sqrt{KH^{-3}}$ then we have

$$\mathcal{R}(K) = \tilde{O}(\sqrt{KH^5} + \sqrt{d^3H^3K}).$$

If the Slater's constant $\rho$ is known, setting $\lambda = \sqrt{KH^{-3}\rho^{-2}}$ then we have

$$\mathcal{R}(K) = \tilde{O}(\frac{\sqrt{KH^5}}{\rho} + \sqrt{d^3 H^3 K}).$$

**Violation analysis**: Let $\pi$ be a strictly feasible policy $V_{g,1}^\pi(x_1^k) \geq b + \rho$. Thus the inequality (3) implies

$$\sum_{k=1}^K V_{r,1}^\pi(x_1^k) - \hat{V}_{r,1}^k(x_1^k) + \rho \sum_{k=1}^K \eta_k - \sum_{k=1}^K \eta_k(\hat{V}_{g,1}^k(x_1^k) - b) \leq H^2(1 + \log|\mathcal{A}|)(\sqrt{K} + \sqrt{\sum_{k=1}^K \eta_k^2}). \tag{55}$$

We also plug (4) such that

$$\sum_{k=1}^K V_{r,1}^\pi(x_1^k) - \hat{V}_{r,1}^k(x_1^k) + \rho \sum_{k=1}^K \eta_k + \lambda \sum_{k=1}^K (b - \hat{V}_{g,1}^k(x_1^k))_+^2 \tag{56}$$

$$\leq H^2(1 + \log|\mathcal{A}|)(\sqrt{K} + \lambda \sqrt{\sum_{k=1}^K (b - \hat{V}_{g,1}^k(x_1^k))_+^2}). \tag{57}$$

Similarly, we add the term of $\lambda H^4(1 + \log|\mathcal{A}|)^2/4$ above and drop the positive terms

$$\lambda \rho \sum_{k=1}^K (b - \hat{V}_{g,1}^k(x_1^k))_+ \leq \sum_{k=1}^K \hat{V}_{r,1}^k(x_1^k) - V_{r,1}^\pi(x_1^k) + H^2(1 + \log|\mathcal{A}|)\sqrt{K} + \frac{\lambda H^4(1 + \log|\mathcal{A}|)^2}{4}. \tag{58}$$

This implies

$$\sum_{k=1}^K (b - \hat{V}_{g,1}^k(x_1^k))_+ \leq \frac{\sum_{k=1}^K \hat{V}_{r,1}^k(x_1^k) - V_{r,1}^\pi(x_1^k)}{\lambda\rho} + \frac{H^2(1 + \log|\mathcal{A}|)\sqrt{K}}{\lambda\rho} + \frac{H^4(1 + \log|\mathcal{A}|)^2}{4\rho}, \tag{59}$$

which finally gives

$$\sum_{k=1}^K (b - V_{g,1}^{\pi_k}(x_1^k))_+ = \sum_{k=1}^K (b - \hat{V}_{g,1}^k(x_1^k) + \hat{V}_{g,1}^k(x_1^k) - V_{g,1}^{\pi_k}(x_1^k))_+$$

$$\leq \frac{HK}{\lambda\rho} + \frac{H^2(1 + \log|\mathcal{A}|)\sqrt{K}}{\lambda\rho} + \frac{H^4(1 + \log|\mathcal{A}|)^2}{\rho} + \tilde{O}(\sqrt{d^3 H^3 K})$$

This completes the regret analysis by invoking Lemma F.5 on the optimistic learning errors $\tilde{O}(\sqrt{d^3 H^3 K})$.

If the Slater's constant $\rho$ is unknown, setting $\lambda = \sqrt{KH^{-3}}$ then we have

$$\mathcal{V}_+(K) = \tilde{O}(\frac{\sqrt{KH^5}}{\rho} + \sqrt{d^3 H^3 K}).$$

If the Slater's constant $\rho$ is known, setting $\lambda = \sqrt{KH^{-3}\rho^{-2}}$ then we have

$$\mathcal{V}_+(K) = \tilde{O}(\sqrt{KH^5} + \sqrt{d^3 H^3 K}).$$

$\square$

## D. Strong duality for CMDP

The following lemma is classical in the optimization literature (Nocedal, 2006) and also demonstrated in the CMDP setting (Paternain et al., 2019; Stradi et al., 2025a).

**Lemma D.1.** *Consider the constrained Markov decision process (CMDP) optimization problem*

$$\max_{\pi} \quad V_{r,1}^{\pi}(x_1), \tag{60}$$

$$s.t. \quad V_{g,1}^{\pi}(x_1) \geq b, \tag{61}$$

*where $V_{r,1}^{\pi}(x_1)$ and $V_{g,1}^{\pi}(x_1)$ denote the expected cumulative reward and constraint return under a policy $\pi$, starting from initial state $x_1$. Assume the value functions are bounded as $V_{r,1}^{\pi}(x_1), V_{g,1}^{\pi}(x_1) \in [0, H]$ for any policy $\pi$. Assume Slater's condition holds: there exists a policy $\pi'$ and a positive constant $\rho > 0$ such that $V_{g,1}^{\pi'}(x_1) \geq b + \rho$. Let $\pi^*$ denote an optimal solution to (60)–(61). Then, for all policies $\pi$,*

$$V_{r,1}^{\pi}(x_1) \; + \; \min_{0 \leq \xi \leq H/\rho} \xi \big(V_{g,1}^{\pi}(x_1) - b\big) \; \leq \; V_{r,1}^{\pi^*}(x_1).$$

*Proof.* Let $\xi$ be the Lagrangian dual associated with the constraint (61). We then introduce the Lagrangian function

$$\mathcal{L}(\pi, \xi) = V_{r,1}^{\pi}(x_1) + \xi \big(V_{g,1}^{\pi}(x_1) - b\big), \quad \xi \geq 0.$$

Under Slater's condition, the strong duality holds for the CMDP in (60)–(61) according to (Paternain et al., 2019). There exists a saddle point $(\pi^*, \xi^*)$ such that

$$\mathcal{L}(\pi, \xi^*) \leq \mathcal{L}(\pi^*, \xi^*) = V_{r,1}^{\pi^*}(x_1), \quad \forall \pi.$$

Therefore, we have

$$V_{r,1}^{\pi}(x_1) + \xi^* \big(V_{g,1}^{\pi}(x_1) - b\big) \leq V_{r,1}^{\pi^*}(x_1), \quad \forall \pi. \tag{62}$$

It remains to upper bound the optimal dual variable $\xi^*$. Let $\pi'$ be the strictly feasible policy satisfying Slater's condition in (62), we have

$$V_{r,1}^{\pi^*}(x_1) = \sup_{\pi} \mathcal{L}(\pi, \xi^*) \geq V_{r,1}^{\pi'}(x_1) + \xi^* \big(V_{g,1}^{\pi'}(x_1) - b\big) \geq \xi^* \rho,$$

where we used $V_{r,1}^{\pi'}(x_1) \geq 0$. Since $V_{r,1}^{\pi^*}(x_1) \leq H$, this implies $\xi^* \leq H/\rho$. Therefore, we have

$$\min_{0 \leq \xi \leq H/\rho} \xi \big(V_{g,1}^{\pi}(x_1) - b\big) \leq \xi^* \big(V_{g,1}^{\pi}(x_1) - b\big), \quad \forall \pi.$$

Combining the inequality with (62) completes the proof. $\qquad\square$

**Lemma D.2** (Restate of Lemma 5.4). *Suppose a policy $\{\pi_k\}_{k=1}^K$ has the weak regret and strong violation are bounded*

$$\mathcal{R}(K) := \sum_{k=1}^{K} \Big(V_{r,1}^{\pi^*}(x_1^k) - V_{r,1}^{\pi_k}(x_1^k)\Big) \; \leq \; R_K,$$

$$\mathcal{V}_+(K) := \sum_{k=1}^{K} \big(b - V_{g,1}^{\pi_k}(x_1^k)\big)_+ \; \leq \; B_K.$$

*Then the strong regret also satisfies*

$$\mathcal{R}_+(K) := \sum_{k=1}^{K} \Big(V_{r,1}^{\pi^*}(x_1^k) - V_{r,1}^{\pi_k}(x_1^k)\Big)_+ \; \leq \; R_K + \frac{H}{\rho} B_K.$$

*Proof.* From Lemma D.1, we have for any policy $\pi$ such that

$$V_{r,1}^{\pi}(x_1) \; + \; \min_{0 \leq \lambda \leq H/\rho} \lambda \big(V_{g,1}^{\pi}(x_1) - b\big) \; \leq \; V_{r,1}^{\pi^*}(x_1).$$

Substituting this bound with $\pi = \pi_k$ yields

$$V_{r,1}^{\pi^*}(x_1^k) - V_{r,1}^{\pi_k}(x_1^k) \; \geq \; \min_{0 \leq \lambda \leq H/\rho} \lambda \big(V_{g,1}^{\pi}(x_1) - b\big) \; \geq \; -\frac{H}{\rho} \big(b - V_{g,1}^{\pi_k}(x_1^k)\big)_+,$$

where the second inequality holds because we have either $V_{g,1}^{\pi^k}(x_1^k) \geq b$ with $\lambda = 0$ or $V_{g,1}^{\pi^k}(x_1^k) < b$ with $\lambda = H/\rho$.

Now using $(x)_+ \leq (x+c)_+$ for any non-negative constants $c \geq 0$, we obtain

$$\left(V_{r,1}^{\pi^*}(x_1^k) - V_{r,1}^{\pi_k}(x_1^k)\right)_+ \leq V_{r,1}^{\pi^*}(x_1^k) - V_{r,1}^{\pi_k}(x_1^k) + \frac{H}{\rho}\left(b - V_{g,1}^{\pi^k}(x_1^k)\right)_+.$$

Summing over $k = 1, \ldots, K$ as follows

$$\sum_{k=1}^{K}\left(V_{r,1}^{\pi^*}(x_1^k) - V_{r,1}^{\pi_k}(x_1^k)\right)_+ \leq \sum_{k=1}^{K} V_{r,1}^{\pi^*}(x_1^k) - V_{r,1}^{\pi^k}(x_1^k) + \frac{H}{\rho}\sum_{k=1}^{K}\left(b - V_{g,1}^{\pi_k}(x_1^k)\right)_+$$

$$\leq R_K + \frac{H}{\rho}B_K$$

where we use the bounds of weak regret and strong constraint violation to complete the proof. $\qquad\square$

## E. Proof of Theorem 5.11

Recall that the guarantees of Theorem 5.5 apply to algorithm APMPO($\lambda, \delta$) with $\delta$ as Slater's constant, yielding bounds for strong regret and strong constraint violation under the $\delta$-slack CMDP in (7). Let $\pi_\delta^*$ denote the optimal policy of the $\delta$-slack CMDP in (7) and let $\pi^k$ be the policy deployed by APMPO($\lambda, \delta$) in episode $k$. We then have

$$\mathcal{R}_+^{\delta\text{-slack}}(K) = \sum_{k=1}^{K}\left(V_{r,1}^{\pi_\delta^*}(x_1^k) - V_{r,1}^{\pi^k}(x_1^k)\right)_+ \leq \tilde{O}\left(\frac{H^2K + 2H^3\sqrt{K} + \lambda H^5}{\lambda\delta^2} + \frac{\sqrt{d^3 H^5 K}}{\delta}\right),$$

$$\mathcal{V}_+^{\delta\text{-slack}}(K) = \sum_{k=1}^{K}\left(b_\delta - V_{g,1}^{\pi^k}(x_1^k)\right)_+ \leq \tilde{O}\left(\frac{HK + 2H^2\sqrt{K} + \lambda H^4}{\lambda\delta} + \sqrt{d^3 H^3 K}\right).$$

The optimal policy $\pi^*$ of the original CMDP in (1) is also feasible for the $\delta$-slack CMDP (7), implying $V_{r,1}^{\pi^*}(x_1^k) \leq V_{r,1}^{\pi_\delta^*}(x_1^k)$ for all $k$. We then have

$$\mathcal{R}_+(K) = \sum_{k=1}^{K}\left(V_{r,1}^{\pi^*}(x_1^k) - V_{r,1}^{\pi^k}(x_1^k)\right)_+$$

$$\leq \sum_{k=1}^{K}\left(V_{r,1}^{\pi_\delta^*}(x_1^k) - V_{r,1}^{\pi^k}(x_1^k)\right)_+$$

$$\leq \tilde{O}\left(\frac{H^2K + 2H^3\sqrt{K} + \lambda H^5}{\lambda\delta^2} + \frac{\sqrt{d^3 H^5 K}}{\delta}\right)$$

$$\leq \tilde{O}\left(H^3 K^{\frac{3}{4}} + 2H^4 K^{\frac{1}{4}} + H^5 K^{\frac{1}{2}} + d^{\frac{3}{2}} H^{\frac{5}{2}} K^{\frac{3}{4}}\right).$$

For strong constraint violation, recall that $b_\delta = b - \delta$. We then have

$$\mathcal{V}_+(K) = \sum_{k=1}^{K}\left(b - V_{g,1}^{\pi^k}(x_1^k)\right)_+$$

$$\leq \sum_{k=1}^{K}\left(b_\delta - V_{g,1}^{\pi^k}(x_1^k)\right)_+ + K\delta$$

$$\leq \tilde{O}\left(\frac{HK + 2H^2\sqrt{K} + \lambda H^4}{\lambda\delta} + \sqrt{d^3 H^3 K} + K\delta\right)$$

$$\leq \tilde{O}\left(K^{\frac{1}{2}} H^2 + H^3 + H^4 K^{\frac{1}{4}} + d^{\frac{3}{2}} H^{\frac{3}{2}} K^{\frac{1}{2}} + K^{\frac{3}{4}}\right).$$

Both last inequalities follow by substituting $\lambda = K^{3/4}H^{-1}$ and $\delta = K^{-1/4}$. All terms are of order at most $\tilde{O}(K^{3/4})$ in their dependence on $K$, up to logarithmic factors and polynomial factors in $H$ and $d$. Suppressing these factors, we obtain the $\tilde{O}(K^{3/4})$ bounds in Theorem 5.11.

# F. Auxiliary Lemmas for Optimistic Learning

Most of the results in this section follow the pioneering paper of optimistic learning for (unconstrained) linear MDP in Jin et al. (2020). We present here for completeness.

## F.1. Linear value functions

**Lemma F.1** (Proposition 2.3 in (Jin et al., 2020)). *For a linear MDP, for any policy $\pi$, there exist weights $\{w_h^\pi\}_{h\in[H]}$ such that for any $(x, a, h) \in \mathcal{S} \times \mathcal{A} \times [H]$,*

$$Q_h^\pi(x, a) = \langle \phi(x, a), w_h^\pi \rangle.$$

*Proof.* The linearity of the action–value functions directly follows from the Bellman equation:

$$Q_h^\pi(x, a) = r(x, a) + \mathbb{E}_{x' \sim P_h(\cdot|x,a)} \left[ V_{u,h+1}^\pi(x') \right]$$
$$= \langle \phi(x, a), \theta_h \rangle + \int_{\mathcal{S}} V_{h+1}^\pi(x') \langle \phi(x, a), d\mu_h(x') \rangle.$$

Therefore, we have

$$Q_h^\pi(x, a) = \langle \phi(x, a), w_h^\pi \rangle \text{ with } w_h^\pi = \theta_h + \int_{\mathcal{S}} V_{h+1}^\pi(x') \, d\mu_h(x').$$

$\square$

## F.2. High probability event under optimistic value iteration

**Lemma F.2.** *Consider a finite-horizon linear MDP satisfying Assumption 3.1 with horizon $H$, feature dimension $d$, and bounded feature map $\|\phi(x, a)\|_2 \leq 1$. Let Algorithm APMPO be run with exploration bonus $\beta = c_\beta \cdot dH\sqrt{\iota}$, where $\iota = \log(2dK^{1.5}H/p)$ and $c_\beta > 0$ is a constant. There exists an absolute constant $C$, independent of $c_\beta$, such that for any fixed $p \in (0, 1)$, if we define the event $\mathcal{E}$ by*

$$\forall (k, h) \in [K] \times [H] : \left\| \sum_{\tau=1}^{k-1} \phi_h^\tau \left( V_{h+1}^k(x_{h+1}^\tau) - P_h V_{h+1}^k(x_h^\tau, a_h^\tau) \right) \right\|_{(\Lambda_h^k)^{-1}} \leq C \cdot dH\sqrt{\chi}, \tag{63}$$

*where $\chi = \log\left( \frac{2(c_\beta+1)dK^{1.5}H}{p} \right)$, then*

$$\mathbb{P}(\mathcal{E}) \geq 1 - \frac{p}{2}.$$

*Proof.* To prove the lemma, we adopt a covering argument together with a self-normalized concentration bound.

**Step 1: $\epsilon$-covering number of the $Q$-function class.** We first consider the $\epsilon$-covering number for the class of value functions. To this end, we begin by computing the $\epsilon$-covering number of individual $Q$-functions, which will be used later to control the covering number of the induced value function class.

We introduce the class of $Q$-functions:

$$\mathcal{Q}_j = \{Q_j \mid Q_j(\cdot, \cdot) = w_j^T \phi(\cdot, \cdot) + \beta_j \|\phi(\cdot, \cdot)\|_{\Lambda^{-1}}\},$$

which is parameterized by $w_j$ and $\Lambda$. Define the distance metric as the $\infty$-norm:

$$\text{dist}(Q_1, Q_2) = \sup_{x,a} |Q_1(x, a) - Q_2(x, a)|.$$

Let $A = \beta^2 \Lambda^{-1}$ for notational simplicity. Since the class $\mathcal{Q}_j$ can be re-parameterized by $(w_j, A)$, we have

$$\text{dist}(Q_1, Q_2) = \sup_{x,a} \left| [w_1^T \phi(x,a) + \sqrt{\phi^T(x,a) A_1 \phi(x,a)}] - [w_2^T \phi(x,a) + \sqrt{\phi^T(x,a) A_2 \phi(x,a)}] \right|$$

$$\leq \sup_{\phi: \|\phi\| \leq 1} \left| (w_1 - w_2)^T \phi \right| + \sup_{\phi: \|\phi\| \leq 1} \left| \sqrt{\phi^T A_1 \phi} - \sqrt{\phi^T A_2 \phi} \right|$$

$$\leq \sup_{\phi: \|\phi\| \leq 1} \left| (w_1 - w_2)^T \phi \right| + \sup_{\phi: \|\phi\| \leq 1} \sqrt{\left| \phi^T (A_1 - A_2) \phi \right|}$$

$$= \|w_1 - w_2\| + \sqrt{\|A_1 - A_2\|}$$

$$\leq \|w_1 - w_2\| + \sqrt{\|A_1 - A_2\|_F},$$

where the second-last inequality follows from the fact that $|\sqrt{x} - \sqrt{y}| \leq \sqrt{|x-y|}$. Here $\|\cdot\|$ and $\|\cdot\|_F$ denote the operator norm and Frobenius norm, respectively.

Let $\mathcal{C}_w^\epsilon$ be an $\epsilon/2$-cover of the set $\{w \in \mathbb{R}^d \mid \|w\| \leq 2H\sqrt{dk}\}$ with respect to the $\ell_2$-norm, and $\mathcal{C}_A^\epsilon$ be an $\epsilon^2/4$-cover of the set $\{A \in \mathbb{R}^{d \times d} \mid \|A\|_F \leq d^{1/2}\beta^2\}$ with respect to the Frobenius norm. According to standard covering number bounds, we have

$$|\mathcal{C}_w^\epsilon| \leq \left( 1 + \frac{8H\sqrt{dk}}{\epsilon} \right)^d, \quad |\mathcal{C}_A^\epsilon| \leq \left( 1 + \frac{8d^{1/2}\beta^2}{\epsilon^2} \right)^{d^2}.$$

For any $Q_j \in \mathcal{Q}_j$, there exists $\tilde{Q}_j$ parameterized by $(w_2, A_2)$ where $w_2 \in \mathcal{C}_w^\epsilon$ and $A_2 \in \mathcal{C}_A^\epsilon$ such that $\text{dist}(Q_j, \tilde{Q}_j) \leq \epsilon$. Hence,

$$N_\epsilon^{Q_j} \leq |\mathcal{C}_w^\epsilon| |\mathcal{C}_A^\epsilon|,$$

and

$$\log N_\epsilon^{Q_j} \leq d \log \left( 1 + \frac{8H\sqrt{dk}}{\epsilon} \right) + d^2 \log \left( 1 + \frac{8d^{1/2}\beta^2}{\epsilon^2} \right).$$

**Step 2: Stability of $F$ and the induced policy.** Recall that $F_{\eta_k}^k = Q_r^k + \eta_k Q_g^k$. Suppose $\text{dist}(Q_r, \tilde{Q}_r) \leq \epsilon'$, $\text{dist}(Q_g, \tilde{Q}_g) \leq \epsilon'$, and $|\eta_k - \tilde{\eta}_k| \leq \epsilon'$. Then

$$\text{dist}(F_{\eta_k}^k, \tilde{F}_{\tilde{\eta}_k}^k) = \sup_{x,a} |(Q_r^k + \eta_k Q_g^k) - (\tilde{Q}_r + \tilde{\eta}_k \tilde{Q}_g)|$$

$$\leq \sup_{x,a} |Q_r^k - \tilde{Q}_r| + \eta_k \sup_{x,a} |Q_g^k - \tilde{Q}_g| + \sup_{x,a} |(\tilde{\eta}_k - \eta_k) Q_g^k|$$

$$\leq \epsilon'(1 + \eta_k) + \epsilon' H$$

$$\leq \epsilon'(1 + 2H),$$

where we use $\eta_k \leq H$ and $|Q_g^k(x,a)| \leq H$.

Recall that the policy is generated by KL-regularized online mirror descent:

$$\pi_{k+1}(\cdot \mid x) = \arg\min_\pi \left\{ \langle F(x, \cdot), \pi_k - \pi \rangle + \alpha_k \text{KL}(\pi \| \pi_k) \right\}. \tag{64}$$

The closed form satisfies

$$\pi_{k+1}(\cdot \mid x) \propto \pi_k(\cdot \mid x) \cdot \exp(\alpha_k F(x, \cdot)),$$

which has the same Lipschitz property as the softmax function. Therefore,

$$\|\pi_{k+1}(\cdot \mid x) - \tilde{\pi}(\cdot \mid x)\|_1 \leq 2\alpha_k \epsilon'(1 + 2H) \leq 2\alpha_K \epsilon'(1 + 2H).$$

**Step 3: Covering number of the value function class.** Define the class of value functions

$$\mathcal{V}_j = \{V_j \mid V_j = \sum_a \pi(a \mid \cdot) Q_j(\cdot, a)\}.$$

For any $x$, we have

$$
\begin{aligned}
|V_j^k(x) - \tilde{V}_j(x)| &= \left| \sum_a \pi_k(a \mid x) Q_j^k(x,a) - \sum_a \tilde{\pi}(a \mid x) \tilde{Q}_j(x,a) \right| \\
&\leq \left| \sum_a \pi_k(a \mid x)(Q_j^k - \tilde{Q}_j) \right| + \left| \sum_a (\pi_k - \tilde{\pi}) \tilde{Q}_j \right| \\
&\leq \epsilon' + \|\pi_k(\cdot \mid x) - \tilde{\pi}(\cdot \mid x)\|_1 \|\tilde{Q}_j(x,\cdot)\|_\infty \\
&\leq \epsilon' + H \cdot 2\alpha_K \epsilon'(1 + 2H).
\end{aligned}
$$

Let $\mathcal{C}_\eta^\epsilon$ be an $\epsilon$-cover of $[0, H]$, then $|\mathcal{C}_\eta^\epsilon| \leq (1 + \frac{H}{\epsilon})$. Set

$$
\epsilon' = \frac{\epsilon}{H2\alpha_K(1 + 2H) + 1}.
$$

Then $\mathrm{dist}(V_j^k, \tilde{V}_j) \leq \epsilon$, and hence

$$
\log N_\epsilon^{V_j} \leq d \log\left(1 + \frac{8H\sqrt{dk}}{\epsilon'}\right) + d^2 \log\left(1 + \frac{8d^{1/2}\beta^2}{\epsilon'^2}\right) + \log\left(1 + \frac{H}{\epsilon'}\right).
$$

**Step 4: Decomposition and concentration.** There exists $\tilde{V}_j$ in the $\epsilon$-cover such that $V_j = \tilde{V}_j + \Delta V$ with $\|\Delta V\|_\infty \leq \epsilon$. Then

$$
\left\| \sum_{\tau=1}^k \phi^\tau (V_j - \mathbb{E}[V_j|\mathcal{F}_{\tau-1}]) \right\|_{(\Lambda^k)^{-1}}^2 \leq 2 \left\| \sum_{\tau=1}^k \phi^\tau (\tilde{V}_j - \mathbb{E}[\tilde{V}_j|\mathcal{F}_{\tau-1}]) \right\|_{(\Lambda^k)^{-1}}^2 + 2 \left\| \sum_{\tau=1}^k \phi^\tau (\Delta V - \mathbb{E}[\Delta V|\mathcal{F}_{\tau-1}]) \right\|_{(\Lambda^k)^{-1}}^2 .
$$

The second term is bounded by $8k^2\epsilon^2$.

For the first term, applying the elliptical potential lemma yields

$$
\begin{aligned}
&\left\| \sum_{\tau=1}^k \phi^\tau (\tilde{V}_j(x_\tau) - \mathbb{E}[\tilde{V}_j(x_\tau) \mid \mathcal{F}_{\tau-1}]) \right\|_{(\Lambda^k)^{-1}}^2 \\
&\leq 2H^2 \left[ d\log(k+1) + d\log\left(1 + \frac{8H\sqrt{dk}}{\epsilon'}\right) + d^2 \log\left(1 + \frac{8d^{\frac{1}{2}}\beta^2}{\epsilon'^2}\right) + \log\left(1 + \frac{H}{\epsilon'}\right) + \log\left(\frac{4}{p}\right) \right]
\end{aligned}
$$

, where $\epsilon' = \frac{\epsilon}{H2\alpha_K(1+2H)+1}$.

**Step 5: Final bound.** Substituting the bound on $N_\epsilon^V$, $\beta = c_\beta dH\sqrt{\iota}$, and choosing $\epsilon = \frac{dH}{k}$, we obtain

$$
\left\| \sum_{\tau=1}^k \phi^\tau (V_j - \mathbb{E}[V_j|\mathcal{F}_{\tau-1}]) \right\|_{(\Lambda^k)^{-1}}^2 \leq C_2 H^2 d^2 \log\left(\frac{4(c_\beta + 1)dK^{1.5}H}{p}\right)
$$

for some absolute constant $C_2$. Taking square roots and applying a union bound over $(k, h)$ yields the desired result. $\qquad \square$

### F.3. Recursive relationship of value functions difference

**Lemma F.3.** *There exists an absolute constant $c_\beta$ such that for $\beta = c_\beta dH\sqrt{\iota}$ where $\iota = \log(2dK^{1.5}H/p)$, and for any fixed policy $\pi$, on the event $\mathcal{E}$ defined in Lemma F.2, we have for all $(x, a, h, k) \in \mathcal{S} \times \mathcal{A} \times [H] \times [K]$ that:*

$$
\langle \phi(x,a), \mathbf{w}_h^k \rangle - Q_h^\pi(x,a) = \mathbb{P}_h(V_{h+1}^k - V_{h+1}^\pi)(x,a) + \Delta_h^k(x,a), \tag{65}
$$

*for some $\Delta_h^k(x,a)$ that satisfies $|\Delta_h^k(x,a)| \leq \beta\sqrt{\phi(x,a)^\top (\Lambda_h^k)^{-1} \phi(x,a)}$.*

*Proof.* By Lemma F.1 and the Bellman equation, for any $(x, a, h) \in \mathcal{S} \times \mathcal{A} \times [H]$, we have:

$$Q_h^\pi(x, a) := \langle \phi(x, a), \mathbf{w}_h^\pi \rangle = (r_h + \mathbb{P}_h V_{h+1}^\pi)(x, a) \tag{66}$$

This gives

$$
\begin{aligned}
\mathbf{w}_h^k - \mathbf{w}_h^\pi &= (\Lambda_h^k)^{-1} \sum_{\tau=1}^{k-1} \phi_h^\tau [r_h^\tau + V_{h+1}^k(x_{h+1}^\tau)] - \mathbf{w}_h^\pi \\
&= (\Lambda_h^k)^{-1} \left\{ -\lambda \mathbf{w}_h^\pi + \sum_{\tau=1}^{k-1} \phi_h^\tau [V_{h+1}^k(x_{h+1}^\tau) - \mathbb{P}_h V_{h+1}^\pi(x_h^\tau, a_h^\tau)] \right\} \\
&= \underbrace{-\lambda (\Lambda_h^k)^{-1} \mathbf{w}_h^\pi}_{\mathbf{q}_1} + \underbrace{(\Lambda_h^k)^{-1} \sum_{\tau=1}^{k-1} \phi_h^\tau [V_{h+1}^k(x_{h+1}^\tau) - \mathbb{P}_h V_{h+1}^k(x_h^\tau, a_h^\tau)]}_{\mathbf{q}_2} \\
&\quad + \underbrace{(\Lambda_h^k)^{-1} \sum_{\tau=1}^{k-1} \phi_h^\tau \widetilde{\mathbb{P}}_h (V_{h+1}^k - V_{h+1}^\pi)(x_h^\tau, a_h^\tau)}_{\mathbf{q}_3}
\end{aligned}
$$

Now, we bound the terms on the right-hand side individually. For the first term,

$$|\langle \phi(x, a), \mathbf{q}_1 \rangle| = |\lambda \langle \phi(x, a), (\Lambda_h^k)^{-1} \mathbf{w}_h^\pi \rangle| \le \sqrt{\lambda} \|\mathbf{w}_h^\pi\| \sqrt{\phi(x, a)^\top (\Lambda_h^k)^{-1} \phi(x, a)}. \tag{67}$$

For the second term, given the event $\mathfrak{E}$ defined in Lemma F.2, we have:

$$|\langle \phi(x, a), \mathbf{q}_2 \rangle| \le c_0 \cdot dH \sqrt{\chi} \sqrt{\phi(x, a)^\top (\Lambda_h^k)^{-1} \phi(x, a)}, \tag{68}$$

for an absolute constant $c_0$ independent of $c_\beta$, and $\chi = \log[2(c_\beta + 1)dT/p]$. For the third term,

$$
\begin{aligned}
\langle \phi(x, a), \mathbf{q}_3 \rangle &= \langle \phi(x, a), (\Lambda_h^k)^{-1} \sum_{\tau=1}^{k-1} \phi_h^\tau \mathbb{P}_h (V_{h+1}^k - V_{h+1}^\pi)(x_h^\tau, a_h^\tau) \rangle \\
&= \langle \phi(x, a), (\Lambda_h^k)^{-1} \sum_{\tau=1}^{k-1} \phi_h^\tau (\phi_h^\tau)^\top \int (V_{h+1}^k - V_{h+1}^\pi)(x') d\boldsymbol{\mu}_h(x') \rangle \\
&= \underbrace{\langle \phi(x, a), \int (V_{h+1}^k - V_{h+1}^\pi)(x') d\boldsymbol{\mu}_h(x') \rangle}_{p_1} - \underbrace{\lambda \langle \phi(x, a), (\Lambda_h^k)^{-1} \int (V_{h+1}^k - V_{h+1}^\pi)(x') d\boldsymbol{\mu}_h(x') \rangle}_{p_2},
\end{aligned}
$$

where by Assumption 3.1, we have

$$p_1 = \mathbb{P}_h (V_{h+1}^k - V_{h+1}^\pi)(x, a), \quad |p_2| \le 2H \sqrt{d\lambda} \sqrt{\phi(x, a)^\top (\Lambda_h^k)^{-1} \phi(x, a)}. \tag{69}$$

Finally, since $\langle \phi(x, a), \mathbf{w}_h^k - \mathbf{w}_h^\pi \rangle = \langle \phi(x, a), \mathbf{q}_1 + \mathbf{q}_2 + \mathbf{q}_3 \rangle$, we have:

$$|\langle \phi(x, a), \mathbf{w}_h^k \rangle - Q_h^\pi(x, a) - \mathbb{P}_h (V_{h+1}^k - V_{h+1}^\pi)(x, a)| \le c' dH \sqrt{\chi} \sqrt{\phi(x, a)^\top (\Lambda_h^k)^{-1} \phi(x, a)}, \tag{70}$$

for an absolute constant $c'$ independent of $c_\beta$. To prove this lemma, we only need to show that there exists a choice of absolute constant $c_\beta$ so that

$$c' \sqrt{\iota + \log(c_\beta + 1)} \le c_\beta \sqrt{\iota} \tag{71}$$

where $\iota = \log(2dK^{1.5}H/p)$. We know $\iota \in [\log 2, \infty]$ by its definition, and $c'$ is an absolute constant $c'$ independent of $c_\beta$. Therefore, we can pick an absolute constant $c_\beta$ which satisfies $c' \sqrt{\log 2 + \log(c_\beta + 1)} \le c_\beta \sqrt{\log 2}$. This choice of $c_\beta$ will make Eq. (71) hold for all $\iota \in [\log 2, \infty)$, which finishes the proof. $\square$

### F.4. Optimistic estimation errors

**Lemma F.4.** *Let* $\Lambda_h^k = \sum_{\tau=1}^{k-1} \phi(x_{\tau,h}, a_{\tau,h})\phi(x_{\tau,h}, a_{\tau,h})^\top + \lambda I$. *Then, for any* $h \in [H]$, *we have:*

$$\sum_{k=1}^K \phi(x_{k,h}, a_{k,h})^\top (\Lambda_h^k)^{-1} \phi(x_{k,h}, a_{k,h}) \leq 2d \log\left(1 + \frac{K}{\lambda d}\right). \tag{72}$$

*Furthermore, the sum of the exploration bonuses is bounded by:*

$$\sum_{k=1}^K \sum_{h=1}^H \min\left\{ \beta\sqrt{\phi(x_{k,h}, a_{k,h})^\top (\Lambda_h^k)^{-1} \phi(x_{k,h}, a_{k,h})}, H \right\} \leq \beta\sqrt{2dHT \log\left(1 + \frac{K}{\lambda d}\right)} + H^2 K. \tag{73}$$

*More precisely, following the standard proof in this paper (using Cauchy-Schwarz and the Elliptical Potential Lemma):*

$$\sum_{k=1}^K \|\phi(x_{k,h}, a_{k,h})\|_{(\Lambda_h^k)^{-1}} \leq \sqrt{K \sum_{k=1}^K \|\phi(x_{k,h}, a_{k,h})\|_{(\Lambda_h^k)^{-1}}^2} \leq \sqrt{2Kd \log\left(1 + \frac{K}{\lambda d}\right)}. \tag{74}$$

*Proof.* The proof relies on the Elliptical Potential Lemma. First, we recall the property of the determinant of the covariance matrix. Let $\Lambda_h^k = \Lambda_h^{k-1} + \phi_{k,h}\phi_{k,h}^\top$, where $\phi_{k,h} = \phi(x_{k,h}, a_{k,h})$. Using the matrix determinant lemma, we have:

$$\det(\Lambda_h^{k+1}) = \det(\Lambda_h^k)(1 + \phi_{k,h}^\top (\Lambda_h^k)^{-1} \phi_{k,h}). \tag{75}$$

Taking the logarithm of both sides:

$$\log \det(\Lambda_h^{k+1}) - \log \det(\Lambda_h^k) = \log(1 + \|\phi_{k,h}\|_{(\Lambda_h^k)^{-1}}^2). \tag{76}$$

Using the inequality $x \leq 2\log(1 + x)$ for $x \in [0, 1]$ (assuming normalized features and appropriate $\lambda$ so the term is small, otherwise the sum is bounded by the horizon $H$), or more standardly, using the fact that for $x \geq 0$, $\log(1 + x) \leq x$, we focus on the upper bound of the sum. The Elliptical Potential Lemma (Lemma D.2 in the paper) states:

$$\sum_{k=1}^K \min\left\{1, \|\phi_{k,h}\|_{(\Lambda_h^k)^{-1}}^2\right\} \leq 2\log\left(\frac{\det(\Lambda_h^{K+1})}{\det(\Lambda_h^1)}\right). \tag{77}$$

Since $\|\phi\| \leq 1$, we have $\det(\Lambda_h^{K+1}) \leq (\lambda + K)^d$. Also $\det(\Lambda_h^1) = \lambda^d$. Thus:

$$\sum_{k=1}^K \|\phi_{k,h}\|_{(\Lambda_h^k)^{-1}}^2 \leq 2d \log\left(1 + \frac{K}{\lambda}\right). \tag{78}$$

To bound the sum of the square roots (which appear in the bonus term), we apply the Cauchy-Schwarz inequality:

$$\sum_{k=1}^K \|\phi_{k,h}\|_{(\Lambda_h^k)^{-1}} \leq \sqrt{K \sum_{k=1}^K \|\phi_{k,h}\|_{(\Lambda_h^k)^{-1}}^2} \tag{79}$$

$$\leq \sqrt{2Kd \log\left(1 + \frac{K}{\lambda}\right)}. \tag{80}$$

Summing over $h = 1, \ldots, H$ gives the final bound used in Theorem 3.1:

$$\sum_{h=1}^H \sum_{k=1}^K \beta\|\phi_{k,h}\|_{(\Lambda_h^k)^{-1}} \leq \beta H\sqrt{2Kd \log\left(1 + \frac{K}{\lambda}\right)} = \tilde{O}(\beta H\sqrt{dK}). \tag{81}$$

Substituting $\beta = \tilde{O}(d\sqrt{H})$, we obtain the total regret scaling of $\tilde{O}(\sqrt{d^3 H^3 K})$. $\qquad\square$

**Lemma F.5** (Value function estimation error bound). *Consider the linear MDP setting with feature dimension $d$ and horizon $H$. Assume the good event $\mathcal{E}$ holds, under which the confidence bounds for the linear value function approximation are valid. Then, with probability at least $1 - \delta$, the cumulative difference between the optimistic value functions and the values induced by the executed policies satisfy*

$$\sum_{k=1}^{K} \left( V_1^k(x_{k,1}) - V_1^{\pi_k}(x_{k,1}) \right) \leq \tilde{O}\left( \sqrt{d^3 H^3 K} \right). \tag{82}$$

*Proof.* We first define a "Good Event". Let $\mathcal{E}$ be the "good event" where the estimated weight vectors $w_h^k$ are close to the true parameters. Specifically, we rely on the concentration of self-normalized processes. Under the good event $\mathcal{E}$, for all $k \in [K]$ and $h \in [H]$, we have:

$$\left| \phi(x,a)^\top w_h^k - \phi(x,a)^\top w_h^* - \mathbb{P}_h(V_{h+1}^k - V_{h+1}^*)(x,a) \right| \leq \beta \sqrt{\phi(x,a)^\top (\Lambda_h^k)^{-1} \phi(x,a)}, \tag{83}$$

where $\beta$ is the bonus scaling parameter chosen in the algorithm (typically $\tilde{O}(d\sqrt{H})$).

We analyze the difference $\delta_h^k = V_h^k(x_{k,h}) - V_h^{\pi_k}(x_{k,h})$. By the definition of the update rule and the Bellman equation, we have:

$$V_h^k(x_{k,h}) - V_h^{\pi_k}(x_{k,h}) = Q_h^k(x_{k,h}, a_{k,h}) - Q_h^{\pi_k}(x_{k,h}, a_{k,h}) \tag{84}$$

$$\leq \phi(x_{k,h}, a_{k,h})^\top w_h^k + \beta \sqrt{\phi_{k,h}^\top (\Lambda_h^k)^{-1} \phi_{k,h}} - (r_h + \mathbb{P}_h V_{h+1}^{\pi_k}) \tag{85}$$

$$= \phi_{k,h}^\top (w_h^k - w_h^*) + \mathbb{P}_h V_{h+1}^* - \mathbb{P}_h V_{h+1}^{\pi_k} + \beta \|\phi_{k,h}\|_{(\Lambda_h^k)^{-1}} \tag{86}$$

$$= \underbrace{\phi_{k,h}^\top (w_h^k - w_h^*) - \mathbb{P}_h(V_{h+1}^k - V_{h+1}^*)}_{\text{Concentration Term}} + \mathbb{P}_h(V_{h+1}^k - V_{h+1}^{\pi_k}) + \beta \|\phi_{k,h}\|_{(\Lambda_h^k)^{-1}}. \tag{87}$$

Using the concentration bound (Good Event), the first term is bounded by the bonus. Thus:

Fix any episode $k$. Under the good event $\mathcal{E}$, the recursive optimism argument yields for all $h \in [H]$:

$$V_h^k(x_{k,h}) - V_h^{\pi_k}(x_{k,h}) \leq 2\beta \|\phi(x_{k,h}, a_{k,h})\|_{(\Lambda_h^k)^{-1}} + \mathbb{P}_h(V_{h+1}^k - V_{h+1}^{\pi_k}), \tag{88}$$

where $\beta = \tilde{O}(d\sqrt{H})$.

Define the martingale difference

$$\xi_h^k = \mathbb{P}_h(V_{h+1}^k - V_{h+1}^{\pi_k}) - \left( V_{h+1}^k(x_{k,h+1}) - V_{h+1}^{\pi_k}(x_{k,h+1}) \right).$$

Unrolling the recursion from $h = 1$ to $H$ gives

$$V_1^k(x_{k,1}) - V_1^{\pi_k}(x_{k,1}) \leq \sum_{h=1}^{H} 2\beta \|\phi(x_{k,h}, a_{k,h})\|_{(\Lambda_h^k)^{-1}} + \sum_{h=1}^{H} \xi_h^k. \tag{89}$$

Summing over episodes $k = 1, \ldots, K$, we obtain

$$\sum_{k=1}^{K} \left( V_1^k(x_{k,1}) - V_1^{\pi_k}(x_{k,1}) \right) \leq 2\beta \sum_{k=1}^{K} \sum_{h=1}^{H} \|\phi(x_{k,h}, a_{k,h})\|_{(\Lambda_h^k)^{-1}} + \sum_{k=1}^{K} \sum_{h=1}^{H} \xi_h^k. \tag{90}$$

The martingale term is bounded via Azuma–Hoeffding:

$$\sum_{k,h} \xi_h^k = \tilde{O}(H\sqrt{K}).$$

For the bonus term, applying the Elliptical Potential Lemma and Cauchy–Schwarz yields

$$\sum_{k=1}^{K} \sum_{h=1}^{H} \|\phi(x_{k,h}, a_{k,h})\|_{(\Lambda_h^k)^{-1}} \leq \tilde{O}(\beta H \sqrt{dK}).$$

Substituting $\beta = \tilde{O}(d\sqrt{H})$ completes the proof:

$$\sum_{k=1}^{K} \left( V_1^k(x_{k,1}) - V_1^{\pi_k}(x_{k,1}) \right) = \tilde{O}(\sqrt{d^3 H^3 K}).$$

$\square$

## G. Simulation Supplements

### G.1. Environment Settings

In our experiments, we follow Kitamura et al. (2025) in building the simulation environment, which includes two different synthetic scenarios.

**Tabular environment.** We construct tabular CMDPs with $|\mathcal{S}| = 5$, $|\mathcal{A}| = 3$ and $H = 4$. For each $h \in [H], x \in \mathcal{S}, a \in \mathcal{A}$, the transition probabilities $P_h(\cdot \mid x, a)$ are sampled independently from $\texttt{Dirichlet}(0.1, \ldots, 0.1)$. Setting the parameter of the Dirichlet distribution to 0.1 makes the transitions of the MDP mostly deterministic but sometimes stochastic. For reward and utility values, $r_h(x, a)$ and $u_h(x, a)$ are independently sampled from $\texttt{Bernoulli}(0.9)$. The initial state $x_1$ is uniformly chosen from $\mathcal{S}$. The utility threshold is defined by $b = 0.6 \max_\pi V_{g,1}^\pi(x_1)$, which is the $60\%$ of the maximum utility.

**Linear environment.** In Linear environment, we set $|\mathcal{S}| = 100, |\mathcal{A}| = 3, H = 4$, and let feature map dimension $d = 5$.

For each $(x, a) \in \mathcal{S} \times \mathcal{A}$, we sample $\phi(x, a) \in \mathbb{R}^d$ independently from $\texttt{Dirichlet}(0.1, \ldots, 0.1)$. Recall Assumption 3.1, for each step $h$, we sample $\mu_h(\cdot) \in \mathbb{R}^d$ independently from $\texttt{Dirichlet}(0.1, \ldots, 0.1)$ and sample $\theta_{r,h}$ and $\theta_{g,h}$ from a uniform distribution over $[0, 1]^d$. Therefore we have

$$r_h(x, a) = \langle \phi(x, a), \theta_{r,h} \rangle, \quad g_h(x, a) = \langle \phi(x, a), \theta_{g,h} \rangle, \quad P_h(\cdot \mid x, a) = \langle \phi(x, a), \mu_h(\cdot) \rangle.$$

The initial state $x_1$ is uniformly chosen from $\mathcal{S}$. The utility threshold is defined by $b = 0.9 \max_\pi V_{g,1}^\pi(x_1)$.

| Algorithm | Tabular | Linear |
|---|---|---|
| Ghosh et al. (2024) | $\beta_r = 1, \beta_g = 1$ | $\beta_r = 5, \beta_g = 1$ |
| Stradi et al. (2025a) | $\beta_r = 1, \beta_g = 0, C_\lambda = 20$ | $\beta_r = 1, \beta_g = 0, C_\lambda = 1$ |
| APMPO(ours) | $\beta_r = 1, \beta_g = 0, \lambda = 10$ | $\beta_r = 1, \beta_g = 0, \lambda = 10$ |

*Table 2.* Hyperparameters for APMPO and baselines in different environments.

### G.2. Baselines

**Implementation of Ghosh et al. (2024).** We use the implementation of Kitamura et al. (2025) for Ghosh et al. (2024), where $\beta_r$ and $\beta_g$ are the UCB scaler.

**Implementation of Stradi et al. (2025a).** Since the method of (Stradi et al., 2025a) is limited to tabular settings, we adapt its core idea of using an upper confidence bound (UCB)-based dual update in place of the computationally expensive search for suitable dual variables in Ghosh et al. (2024) to extend to linear approximation settings. In our implementation, $\beta_r$ and $\beta_g$ denote the UCB scaling parameters for the reward and constraint cost, respectively, and $C_\lambda$ is the dual variable value corresponding to the condition that the optimistic value estimate falls below a prescribed threshold.

Additionally, we also report the performance of a uniform policy defined by $\pi_h(\cdot \mid s) = \frac{1}{A}$ for all $h, s$, to highlight the sublinear regret of our algorithm.

We list the hyperparameters for APMPO and baselines in Table 2.

