# OpenReview forum: "Towards Achieving Optimal Strong Regret and Constraint Violation via Computationally Efficient Model-free RL"
_ICML.cc/2026/Conference — ICML 2026 regular_

### Official Review · Reviewer_eHBj · 2026-02-13

**Soundness:** 2
**Presentation:** 3
**Significance:** 2
**Originality:** 3
**Overall Recommendation:** 4
**Confidence:** 4

**Summary:**

This paper proposes a model-free algorithm for learning episodic constrained Markov decision processes with linear function approximation, and provides theoretical guarantees for the so-called strong reward regret and constraint violation. Compared with prior work, the proposed algorithm doesn't rely on knowledge of Slater's condition for setting the parameters of the algorithm, yet still achieves a regret of order $\sqrt{K}$.

**Compliance With Llm Reviewing Policy:**

Affirmed.

**Final Justification:**

The authors solve my concerns.

**Key Questions For Authors:**

1. See my first comments on the major issue.
2. In Theorem 5.9, the authors study results beyond Slater’s condition. Since Slater’s condition is not assumed, the well-definedness of the optimal policy $\pi^*$ requires the existence of at least one feasible policy, which may lie on the boundary of the feasible region. The authors should explicitly state this assumption.
3. What is the definition of (reward) regret and violation regret used for plotting results in the paper, as the authors used different notions for regrets?

I would be happy to raise my scores if my concerns are addressed.

**Limitations:**

The authors should have detail discussion on works about safe exploration, both model-free or model-based approaches, as the regret there directly implies the regret considered in this paper.

1. Liu, Xin, et al. "An efficient pessimistic-optimistic algorithm for stochastic linear bandits with general constraints." Advances in Neural Information Processing Systems 34 (2021): 24075-24086.
2. Bura, Archana, et al. "DOPE: Doubly optimistic and pessimistic exploration for safe reinforcement learning." Advances in neural information processing systems 35 (2022): 1047-1059.
3. Yu, Kihyun, et al. "Improved Regret Bound for Safe Reinforcement Learning via Tighter Cost Pessimism and Reward Optimism." arXiv preprint arXiv:2410.10158 (2024).
3. Ni, Tingting, et al."A safe exploration approach to constrained Markov decision processes." arXiv preprint arXiv:2312.00561 (2023).

**Strengths And Weaknesses:**

Strengths: The experiments show proposed algorithm significantly outperforms the strong baselines.

Major issue: In the proof of Lemma 5.5, the authors made an error in Inequality (16), which plays an important role in proving Lemma 5.5. The correct inequality should be

$ <l_h^k, \pi_{k+1,h}-\pi_{k,h}>-\alpha_k KL(\pi_{k+1,h}||\pi_{k,h})\le \frac{\|l_h^k\|_{\infty}}{2\alpha_k}$

instead of

   $ <l_h^k, \pi_{k+1,h}-\pi_{k,h}>+\alpha_k KL(\pi_{k+1,h}||\pi_{k,h})\le \frac{\|l_h^k\|_{\infty}}{2\alpha_k}.$

This inequality affects the entire result of Lemma 5.5, and therefore also affects the main result of the paper, Theorem 5.2.

Minor issue: For Theorem 5.2, Lemma 5.6, and Theorem 5.9, what is the dependence of the regret bound on the confidence parameter $\delta$? The results are stated to hold for any $\delta\in(0,1)$, but the final bound does not appear to depend on $\delta$ at all. This is difficult to justify. For example, if $\delta$ is set exponentially small, such as $\delta = e^{-K}$, the final regret bound should depend on $\delta$, even under big-O notation.



Typos and clarity issues:

1. The problem formulation assumes a fixed initial state $x_1$. However, in the presentation of the algorithm (left side of line 202) and the definition of regret (right side of line 252), the authors use $x_1^k$ as the initial state at iteration $k$. For better clarity and consistency, it should be explicitly stated that $x_1 = x_1^k$.

2. In the statement of Theorem 5.2, it should refer to (reward) regret instead of strong (reward) regret.

3. The statement of Lemma 5.3 requires Slater’s condition to hold, since it is used in Lemma D.1, which is in turn used to derive Lemma 5.3.

4. In the statement of Theorem 5.2, it would be better to explicitly specify the choice of the penalty parameter $\lambda$, as it depends on whether Slater’s constant $\rho$ is known.

---

> ### Author Rebuttal · Authors · 2026-03-31
>
> We sincerely appreciate the reviewer's time and constructive feedback on our paper. We also apologize for a typo in Lemma 5.5, which may have caused some misunderstanding. We will address the concerns below.
>
> **Typo in proof of Lemma 5.5.** We thank the reviewer for carefully checking our proofs and pointing out the mismatch. There is a typo when invoking the Fenchel–Young inequality (14) for the negative entropy mirror map. Specifically, we need to set $p = \pi_{k+1, h}(\cdot|x)$, $q = \pi_{k, h}(\cdot|x)$, and $\underline{v = -\ell_h^k(x, \cdot)}$ to obtain the following bound:
> $$
> \langle \underline{-\ell\_h^k(x, \cdot)}, \pi\_{k+1, h}(\cdot|x) - \pi\_{k, h}(\cdot|x) \rangle - \alpha\_k \mathrm{KL}(\pi\_{k+1, h}(\cdot|x) \\| \pi\_{k, h}(\cdot|x)) \leq \frac{\\|\ell\_h^k(x, \cdot)\\|\_\infty^2}{2\alpha\_k}.
> $$
> We then add this inequality to inequality (12) in the paper, such that the following inequality holds
> $$\bigl\langle\ell\_h^k(x,\cdot),\pi\_{k,h}(\cdot|x)-\pi\_h(\cdot|x)\bigr\rangle\leq\alpha\_k\mathrm{KL}\left(\pi\_h(\cdot|x)\\|\pi_{k,h}(\cdot|x)\right)-\alpha\_k\mathrm{KL}\left(\pi\_h(\cdot|x)\\|\pi\_{k+1,h}(\cdot|x)\right) +\frac{\\|\ell_h^k(x,\cdot)\\|\_\infty^2}{2\alpha\_k},
> $$
> which is exactly the inequality (without summation $\sum_{k=1}^K$) in (17)-(18) in the paper. Therefore, the results of Lemma 5.5 and Theorem 5.2 remain valid. We apologize for the mismatch again and have fixed this typo in the revision.
>
>
> **The discussion on $\delta$.** We would like to clarify that the slack parameter $\delta$ is only introduced in Section 5.3 (Theorem 5.9) to handle the more challenging setting without Slater's condition. This is a design parameter in our algorithm and analysis rather than a problem-dependent factor.  In other words, $\delta$ is inherently a tunable (or trade-off) parameter, and we set $\delta=K^{-\frac{1}{4}}$ as in Theorem 5.9.
>
> **Clarification on the feasible policy without Slater’s condition.** We thank the reviewer for this great point. It is certainly necessary that we need to assume the existence of at least one feasible policy. We have explicitly added this assumption in the revision.
>
> **The regret metric in the figures.** Figure 1 depicts the strong regret constraint violation, and we have explicitly clarified it in the revision.
>
> **Related work/Limitations.** We thank the reviewer for introducing these great papers on safe exploration and would definitely incorporate them in our revision. However, we want to emphasize that our paper differs quite significantly from these papers with respect to the regret metric, algorithm design, and technique assumptions. If we read them correctly, we studied model-free RL with function approximation under strong regret, whereas all these papers focused on standard (soft) regret. Liu et al. (2021) considered the bandit setting; Bura et al. (2022) and Yu et al. (2024) studied model-based RL in tabular settings; Bura et al. (2022), Yu et al. (2024), and Ni et al. (2023) all assumed that a strictly safe base policy is known, thereby enabling safe exploration. We will discuss and compare them in detail in the revision.
>
>
> We hope that our response addresses the reviewer's concerns and that the reviewer can re-evaluate our work and raise the scores. Please let us know if there are any further comments, and we will try our best to address them.

---

> > ### Author Rebuttal · Reviewer_eHBj · 2026-03-31
> >
> > Thanks for your rebuttal.
> >
> > 1. My concerns regarding the proof of Lemma 5.5, the clarification of the feasible policy without Slater’s condition, and the regret metric used in the figures have been well addressed.
> >
> > 2. Regarding the questions on related work, the works I listed focus on safe exploration, which implies that the strong regret considered in your paper is zero during the learning process, rather than the standard (soft) regret.
> >
> > 3. Sorry for the confusion. Here, I meant the discussion on the confidence parameter \( p \), not the slack parameter \( \delta \).
> >
> > 4. I have questions regarding Inequality (4) in the main paper part, where the authors write
> >
> > $$
> > \eta_k(b - \hat V) = \frac{\eta_k^2}{\lambda}.
> > $$
> >
> > Considering the case where $b - \hat V \le 0 $, we have $\eta_k(b - \hat V) = 0 $ since $\eta_k = 0$. Therefore, the correct inequality should be
> >
> > $$
> > \eta_k(b - \hat V) \le \frac{\eta_k^2}{\lambda}.
> > $$
> >
> > Consequently, this would also affect the rest of the proof.

---

> > > ### Author Response · Authors · 2026-04-02
> > >
> > > We are delighted to learn that your major concern has been addressed. We sincerely appreciate your time and willingness to discuss this further. Regarding the remaining questions, we would like to offer the following clarifications.
> > >
> > > **Further discussion on related work.** We appreciate your sharp insight regarding these related works. We would like to clarify that our objective is fundamentally different from the safe exploration setting (Liu et al., 2021; Bura et al., 2022; Yu et al., 2024; Ni et al., 2023).
> > > First, we note that Liu et al. 2021 still considered soft cumulative constraint violation (though it could be zero after some period, but it could introduce large or even linear hard violation). The other three papers indeed can do safe exploration during learning; however, they all rely on the availability of a strictly safe baseline policy (or a known safe set).  As noted in Ni et al. (2023), “it is natural to consider this assumption, since without a safe initial policy, safe exploration cannot be satisfied.”
> > >
> > > In contrast, our work does not assume access to such a strictly safe baseline policy, or even its existence (i.e., it is Slater’s condition-free), and instead operates in a more general model-free setting with function approximation. As a result, guaranteeing zero constraint violation at all times is not possible without sacrificing learning efficiency or introducing strong prior safety knowledge. Our notion of strong regret and violation, therefore, captures the inherent trade-off between exploration and constraint satisfaction in this more challenging setting.
> > >
> > > We will incorporate this discussion to clarify the distinction, as suggested by the reviewer.
> > >
> > > **The clarification on $p$.**
> > > The dependence on (p) in our performance bounds is logarithmic, i.e., an $O(\log(1/p))$ term. Notably, $p$ is also a design parameter in our analysis and is typically chosen to be polynomially small, e.g., $p = O(1/K)$, where its contribution is absorbed into the $\tilde{O}(\cdot)$ notation. As suggested, we have made this explicit in the revised version.
> > >
> > > **Equation (4) is correct.** We thank the reviewer for pointing this out. We note equation (4) is correct due to the fact
> > > $x\cdot\max\\{x, 0\\}=(\max\\{x, 0\\})^2$.  We clarify the confusion in the case $b - \hat V \le 0 $. Here, we have $\eta_k = 0$, therefore,  $\eta_k( b -\hat{V}_{g, 1}^k(x_1^k))=\frac{\eta_k^2}{\lambda}=0$.
> > >
> > > We thank the reviewer once again for the detailed comments, which have definitely helped improve the quality of our paper. We sincerely hope our response addresses your major concerns and that you will consider reevaluating our work. Please let us know if you have any further comments, and we will do our best to address them.

---

### Official Review · Reviewer_6BPx · 2026-02-16

**Soundness:** 3
**Presentation:** 2
**Significance:** 3
**Originality:** 3
**Overall Recommendation:** 4
**Confidence:** 4

**Summary:**

The authors study linear constrained stochastic CMDPs. In such a setting, they provide the first model-free algorithm capable of attaining $\tilde{O}(\sqrt K)$ strong regret and strong violation, without the knowledge of the Slater’s parameter. Without Slater’s condition, they attain $\tilde{O}(K^{3/4})$ strong regret and violation.

**Compliance With Llm Reviewing Policy:**

Affirmed.

**Final Justification:**

I will keep my positive evaluation of the paper. As far as I see, the technical problems have been resolved and the other Reviewers raised their evaluation, thus, I will support acceptance of the paper.

**Key Questions For Authors:**

See Weaknesses

**Limitations:**

yes

**Strengths And Weaknesses:**

**Strengths**

Overall, I find the results provided in the paper valuable. Specifically, I believe that extending the state-of-the-arts results on tabular CMDPs to linear one it is of great interest.

**Weaknesses**

The main weakness of the paper is the lack of a precise discussion on the improvement with respect to the state-of-the-art results in the CMDP literature. Specifically, as I specified before, this paper provides interesting results from an algorithmic side, developing the first model-free procedure for CMDPs, thus tailored for CMDP with infinite state spaces. Nonetheless, I believe that some claims provided in this works are not fully precise. For instance, while removing the Slater’s knowledge assumption may be of interest, it is important to underline that, in stochastic CMDPs, $\rho$  can be estimated in **constant** time employing the approach of [1]. Thus, any algorithm which requires the knowledge of $\rho$ can be then employed after a constant number of episodes. Additionally, while it is good to remove the Slater’s assumption, $K^{3/4}$ regret and violation can be easily attained in by clipping $\rho$ to $K^{-1/4}$. This technique is pretty standard (e.g., see [2]), and it can be easily deployed in many algorithms such as the one in (Stradi et al., 2025)

To conclude, I lean towards the acceptance of the paper. Nonetheless, I believe that a precise discussion on the point specified before should be included in the final version of the paper.

[1] “Learning Adversarial MDPs with Stochastic Hard Constraints”, ICML 2025

[2] “A Unifying Framework for Online Optimization with Long-Term Constraints”, NeurIPS 2022

---

> ### Author Rebuttal · Authors · 2026-03-31
>
> We sincerely thank the reviewer for the insightful feedback and for recognizing our contributions. We will incorporate a more detailed discussion of the two great related work [1] and [2] in the revision.
>
> As the reviewer noted, our work provides the first model-free approach for CMDPs with linear function approximation under strong performance guarantees, which differs fundamentally from [1] and [2]. Following the reviewer’s suggestion, we will clarify these differences, particularly regarding the role of Slater’s condition and Slater's constants. For example, both [1] and [2] involve a dedicated stage to estimate $\rho$, whereas our approach is a one-stage, adaptive method that does not require prior knowledge of $\rho$. We also note that techniques from [1] can potentially be incorporated into our framework. As highlighted in Table 1, when $\rho$ is known, our method achieves even stronger guarantees.
>
> We will include a more precise discussion of these related works in the revision. Please let us know if there are any further comments, and we will try our best to address them.

---

> > ### Author Rebuttal · Reviewer_6BPx · 2026-04-01
> >
> > I would like to thank the Authors for their responses. A precise discussion on the points that I raised should be sufficient, thus I will keep my positive evaluation. In the final discussion with the other Reviewers, I will keep into account if their doubts on the proofs have been adequately resolved.

---

> > > ### Author Response · Authors · 2026-04-02
> > >
> > > We truly appreciate your time and engagement. We will add a detailed discussion as you suggested. We are eager to work with you and the other reviewers to get a positive convergence within the next few days. Please let us know if you have any further comments or concerns, and we would be happy to address them before the discussion period ends.

---

### Official Review · Reviewer_CXiT · 2026-03-08

**Soundness:** 2
**Presentation:** 3
**Significance:** 3
**Originality:** 2
**Overall Recommendation:** 4
**Confidence:** 3

**Summary:**

This work studies constrained linear MDP, where the agent seeks to minimize the regret and at the same time need to minimize the constraint violation. The paper proposes a adaptive penalty matching policy optimization algorithm and proves that it can achieve sub-linear regret and constraint violation.

**Compliance With Llm Reviewing Policy:**

Affirmed.

**Final Justification:**

My concerns about the technical part has been resolved. As long as the work will use the new version of the proof, I believe it is correct.

**Key Questions For Authors:**

Please see the weakness part, which is my main question. How to address this inconsistency?

**Limitations:**

Yes.

**Strengths And Weaknesses:**

Strength:
1. The problem is interesting and important.
2. The algorithm may provide insights to existing literature.

Weakness:
I believe there is a fundamental issue in the proof. The paper directly used self-normalizing bound (Lemma F.2) in Jin et al., 2020 which is established for greedy/max policy, i.e. $V(s) = \max_a w^\top\phi(s,a) + \beta\\|\phi(s,a)\\|\_{A}$. In other words, the values only depend on $w, A, \beta$, and union bound and covering number are valid. However, this work uses $V(s) = \sum_{a}\pi^k(a|s)Q(s,a)$, with learned policy $\pi^k$, union bound and covering number may fail on this policy class.

---

> ### Author Rebuttal · Authors · 2026-03-31
>
> We are grateful to the reviewer for pointing out the subtle inconsistency in invoking the "self-normalizing bound". We believe this issue can be readily addressed by following the methodology of Ghosh et al. (2022) and the fixed version only introduce/add a small item $O(\log \alpha_k)=O(\log K)$ term ($\alpha_k$ is the smooth factor) in the "self-normalizing bound" that will not affect the order in the main results.
>
> The key intuition is that the exponentially weighted policy serves as a smooth approximation of the greedy policy. Consequently, the covering number of the value function class can be bounded by that of the Q-value function class, which corresponds to a standard linear function class.
>
> We present the fixed lemma and outline the key steps of the proof below due to the limited space.
>
> **Fixed Lemma F.2.** There exists a constant $C$ such that for any fixed $p\in(0, 1)$, if we let $\mathcal{E}$ be the event that
> $$
>     \left\\|\sum\_{\tau = 1}^{k-1}\phi^\tau\_h(\hat{V}\_{j,h+1}^k(x\_{h+1}^\tau)-P\_h\hat{V}\_{j,h+1}(x\_h^\tau, a\_h^\tau))\right\\|\_{(\Lambda\_h^k)^{-1}}\leq CdH\sqrt{\chi}
> $$
> for all $j\in\{r, g\}$, with $\chi=\log(4(c_\beta+1))dK^{1.5}H/p)$, then $\Pr(\mathcal{E})\geq1-p/2$.
>
> **Proof Outline.**
> We first note that our target term above (in Lemma F.2.) can be bounded by
> $$
> 2 \left\\|
> \sum\_{\tau=1}^k \phi^\top \big( \tilde V\_j(x_\tau) - \mathbb{E}[\tilde V\_j(x_\tau)\mid \mathcal{F}\_{\tau-1}] \big)
> \right\\|^2\_{(\Lambda^k)^{-1}} \nonumber  + 2 \left\\|
> \sum\_{\tau=1}^k \phi^\top \big( \Delta V(x\_\tau) - \mathbb{E}[\Delta V(x\_\tau)\mid \mathcal{F}\_{\tau-1}] \big)
> \right\\|^2\_{(\Lambda^k)^{-1}},
> $$
> where $\tilde V_j$ in the $\epsilon$-covering for $\mathcal{V}\_j$ such that
> $\hat V_j(x) = \tilde V_j(x) + \Delta V(x), \forall x$ with $\sup_x \Delta V(x) \le \epsilon$.
> The second term is readily bounded by $O(\epsilon^2)$, so we focus on the first term.
> To apply the self-normalized concentration (elliptical lemma), it suffices to control the covering number of the value function class $\mathcal{V}\_j$.
> We construct the covering as follows. For any $(Q_r^k, Q_g^k, \eta_k)$, there exist
> $\tilde Q\_r \in \mathcal{Q}\_r$, $\tilde{Q}\_g \in \mathcal{Q}\_g$, and $\tilde \eta \in [0,\lambda]$ such that
> $$
> \\|Q\_r^k - \tilde Q\_r\\|\_\infty \le \epsilon, \quad
> \\|Q\_g^k - \tilde Q\_g\\|\_\infty \le \epsilon, \quad
> |\eta\_k - \tilde \eta| \le \epsilon,
> $$
> By the smoothness of the softmax operator, the induced policies satisfy
> $$
> \\|\pi\_k(\cdot \mid x) - \tilde \pi(\cdot \mid x)\\|\_1  = O(\alpha\_k \epsilon).
> $$
> Using the decomposition
> \begin{align*}
> |V_j^k(x) - \tilde V_j(x)|
> \le
> \left| \sum_a \pi_k(a\mid x)\big(Q_j^k(x,a) - \tilde Q_j(x,a)\big) \right|  +
> \left| \sum_a \big(\pi_k(a\mid x) - \tilde \pi(a\mid x)\big)\tilde Q_j(x,a) \right|,
> \end{align*}
> and the boundedness of $Q_j$, we obtain
> $\\|V_j^k - \tilde V_j\\|_\infty = O(\alpha_k \epsilon).$
> Therefore, the covering number of $\mathcal{V}_j$ can be controlled by that of $(Q_r, Q_g, \eta)$, yielding an additional term $O(\log (\alpha_K/\epsilon))$, which is $O(\log K)$ and consistent with the previous bound. We apologize for this mismatch again and will provide the complete proof in the revision.
>
> Besides, we also want to thank the reviewer for appreciating our contribution on the design of adaptive penalty matching. We are also very excited about these results that can advance the long-standing problems of CMDP with strong regret and constraint violation guarantees.
>
> We hope that our response addresses the reviewer's concerns and that the reviewer will re-evaluate our work. We are ready to address any further comments.

---

> > ### Author Rebuttal · Reviewer_CXiT · 2026-04-02
> >
> > Thanks for the rebuttal. I have no problems with this result and its proof. Please update it in the revision. I have also decided to raise my score.

---

> > > ### Author Response · Authors · 2026-04-02
> > >
> > > Thank you so much for reconsidering our work and raising your rating! We appreciate your thoughtful comments and constructive suggestions, which have helped us improve the clarity and quality of the paper. We will update it in the revision following your suggestions. Please don't hesitate to let us know if you have any other questions/comments. Thanks!

---

### Official Review · Reviewer_ziHx · 2026-03-13

**Soundness:** 3
**Presentation:** 3
**Significance:** 2
**Originality:** 3
**Overall Recommendation:** 4
**Confidence:** 4

**Summary:**

This paper studies episodic CMDPs with linear function approximation, with the goal for strong regret and strong constraint violation guarantees with a model-free algorithm. The paper proposes APMPO, a model-free policy optimization method with an adaptive violation-aware penalty, and proves a near-optimal $\tilde{O}\sqrt{K}$ strong regret and strong violation under Slater's condition, and further proves a $\tilde{O}(K^{3/4})$ strong regret and strong violation without Slater's condition.

**Compliance With Llm Reviewing Policy:**

Affirmed.

**Final Justification:**

During the rebuttal period, the authors answered my questions and addressed most of my concerns. Therefore, I keep my positive rating for this paper as weak accept.

**Key Questions For Authors:**

1. Following up on the weakness: what is the suboptimality and feasibility of the final returned policy w.r.t. K? Is the single final returned policy $\pi^K$?
2. Could this result be converted to PAC-style sample complexity bounds?
3. Do the authors think the $\tilde{O}(K^{3/4})$ bound without Slater's condition is optimal, or maybe $\tilde{O}(\sqrt{K})$ is still possible in this setting?
4. The $K$-dependence is strong, but how tight is the dependence on $H$ and $d$?
5. Conceptually, why is the adaptive penalty matching approach better than a primal-dual update for strong regret/violations?

**Limitations:**

yes

**Strengths And Weaknesses:**

Strengths:
- This paper tackles an important problem: strong regret and strong violation in model-free CMDPs with function approximation.
- The results are strong. The paper provides near-optimal bounds under Slater's condition.
- The algorithmic design is novel. The paper proposes a violation-adaptive penalties + mirror descent update.
- The paper is well-written and positions itself clearly against prior work on strong metrics of CMDPs.

Weaknesses:
- The resulting bounds of strong regret/violations include fairly large dependence on horizons or feature dimensions, so the actual sharpness of the bounds is not fully clear.
- The guarantees are for the online learning process, and the paper would benefit from discussing the sub-optimality and feasibility of the final output policies as a function of K. In safe RL, we care also about the feasibility of the output policy, and not just the learning process itself.

---

> ### Author Rebuttal · Authors · 2026-03-31
>
> We sincerely appreciate the reviewer's time and constructive feedback. We address your concern below.
>
> **Dependence on $H$ and $d$.**  We thank the reviewer for this insightful observation and clarify as follows:
>
> +  **Dependence on feature dimension $d$:** We agree that the $O(d^{3/2})-$dependence is likely to be optimized as the minimax bound of the bandit setting is $O(d).$ This might requires a tighter confidence radius in establishing self-normalizing bound (e.g., from $O(d)$ to $O(\sqrt{d})$ dependence). Specifically, we believe adopting a Bernstein-type UCB instead of a Hoeffding-type bound might achieve an $O(d)$ rate, thereby matching the lower bound for linear MDP [1]. We will further explore its performance in our methods.
>
> +  **Dependence on horizon $H$:** As discussed above, applying a tighter confidence radius, such as Bernstein-type bounds, might also help reduce one $\sqrt{H}$ factor (Jin et al., 2020 in the paper). However, the dependence on $H$ mainly stems from enforcing both strong (non-canceling) regret and constraint violation guarantees. Unlike standard (weak) metrics, strong metrics penalize per-episode errors without cancellation, which inherently introduces additional multiplicative factors in the analysis (e.g., via the reduction from weak to strong guarantees in Lemma 5.3).
>     We conjecture that this dependence may appear intrinsic to strong guarantees, and improving the horizon dependence remains a challenging and largely open direction.
>
>
> **PAC-style sample complexity bounds and final returned policy guarantee.**  If we understand correctly, the reviewer's first two questions focus on the theoretical guarantees of the final returned policy. We believe our results can be directly translated to a sample complexity guarantee (or PAC guarantee) by using the classical online-to-batch techniques as in Section 3.1 of the great paper [2].
> Specifically, by Theorem 5.2., we can learn an $\epsilon$-optimal policy $\pi_\text{out}$ which satisfies $V^*\_{r,1}(x\_1)-V^{\pi\_\text{out}}\_{r,1}(x_1)\leq\epsilon$ and $b - V^{\pi\_\text{out}}\_{g,1}(x\_1) \leq \epsilon$ using $\tilde{O}(\max\\{\frac{H^6}{\rho^2\epsilon^2}, \frac{d^3H^4}{\epsilon^2}\\})$ samples. Here, the final "returned" policy $\pi_\text{out}$ is obtained after running APMPO for $\tilde{O}(\max\\{\frac{H^5}{\rho^2\epsilon^2}, \frac{d^3H^3}{\epsilon^2}\\})$ episodes (where each episode consists of $H$ steps/samples), and then selecting policy $\pi_k$ with probability $1/K$ for any $k\in[K]$.
>
> **Optimality without Slater's condition.**
> We are grateful to the reviewer for this important question. To the best of our knowledge, the analysis of strong regret and constraint violation without Slater’s condition is still not fully explored. The $\tilde{O}(K^{3/4})$ rate currently represents the best known guarantee in this setting, even for the constrained bandit setting (e.g., [3]). Though we also conjecture that $O(\sqrt{K})$ bounds are likely the right and best order, it could be notoriously challenging and is a large open problem in this field.
>
> **Advantage of adaptive penalty matching approach.** Unlike traditional Primal-Dual methods, which typically regulate soft violations through dual variable updates, our approach directly enforces hard constraints. Primal-Dual methods often suffer from oscillatory behavior and struggle to guarantee instantaneous safety. In contrast, APMPO utilizes an adaptive, episode-wise penalty derived from observed data. By relying solely on the immediately preceding time step, our design facilitates rapid recovery from potential violations. This eliminates the instability of dual updates and ensures a more direct and responsive enforcement of hard constraints.
>
> [1] He, J., Zhou, D., & Gu, Q. Nearly optimal regret for learning adversarial mdps with linear function approximation. In arXiv preprint, 2021.
>
> [2] Jin, C., Allen-Zhu, Z., Bubeck, S., & Jordan, M. I. Is Q-learning provably efficient?. In NeurIPS, 2018.
>
> [3] Slivkins, A., Zhou, X., Sankararaman, K. A., & Foster, D. J. Contextual bandits with packing and covering constraints. In JMLR, 2024.

---

> > ### Author Rebuttal · Reviewer_ziHx · 2026-04-02
> >
> > I thank the authors for the response. Most of my questions are resolved, and I keep my positive score.

---

> > > ### Author Response · Authors · 2026-04-03
> > >
> > > Thanks a lot for your time in helping us improve our paper. Much appreciated! Please let us know if you have any follow-up questions. We will be happy to answer them.

---

### Decision · Program_Chairs · 2026-04-30

**Decision:**

Accept (regular)

**Comment:**

**Summary:** This paper studies episodic CMDPs with linear function approximation, with the goal for strong regret and strong constraint violation guarantees with a model-free algorithm. The paper proposes APMPO, a model-free policy optimization method with an adaptive violation-aware penalty, and proves a near-optimal $O(\sqrt{K})$ strong regret and strong violation under Slater's condition, and further proves a $O(K^{3/4})$ strong regret and strong violation without Slater's condition.

**Strengths:**
- Well-written paper that tackles an important problem: strong regret and strong violation in model-free CMDPs with function approximation.
- Novel algorithm design and strong theoretical guarantees.

**Weaknesses:**
- Lack of lower bounds justifying the dependence on $K$ and horizon $H$ and dimension $d$.
- Missing references (see below)

**Decision and Suggested Changes:** The reviewers unanimously agree that the paper's contributions merit acceptance.  Addressing the following concerns will help strengthen the current version of the paper:
- Comments about the tightness of the dependence on $K, H, d$ and conjecture about the lower-bound (Rev. ziHx)
- Adding the fixed Lemma F.2. (Rev. CXiT)
- Clarification about the estimation of the Slater constant, and the corresponding comparison to the relevant papers (Rev. 6BPx)
- Fix the typo in Lemma 5.5 (Rev. eHBj)
- Add the discussion about $\delta$ and the existence of a feasible policy (Rev. eHBj)
- Comparison to related work (Rev. eHBj)